# Guadecitabine plus ipilimumab in unresectable melanoma: five-year follow-up and integrated multi-omic analysis in the phase 1b NIBIT-M4 trial

Association with hypomethylating agents is a promising strategy to improve the efficacy of immune checkpoint inhibitors-based therapy. The NIBIT-M4 was a phase Ib, dose-escalation trial in patients with advanced melanoma of the hypomethylating agent guadecitabine combined with the anti-CTLA-4 antibody ipilimumab that followed a traditional 3 + 3 design (NCT02608437). Patients received guadecitabine 30, 45 or 60 mg/m$^2$/day subcutaneously on days 1 to 5 every 3 weeks starting on week 0 for a total of four cycles, and ipilimumab 3 mg/kg intravenously starting on day 1 of week 1 every 3 weeks for a total of four cycles. Primary outcomes of safety, tolerability, and maximum tolerated dose of treatment were previously reported. Here we report the 5-year clinical outcome for the secondary endpoints of overall survival, progression free survival, and duration of response, and an exploratory integrated multi-omics analysis on pre- and on-treatment tumor biopsies. With a minimum follow-up of 45 months, the 5-year overall survival rate was 28.9% and the median duration of response was 20.6 months. Re-expression of immunomodulatory endogenous retroviruses and of other repetitive elements, and a mechanistic signature of guadecitabine are associated with response. Integration of a genetic immunoediting index with an adaptive immunity signature stratifies patients/lesions into four distinct subsets and discriminates 5-year overall survival and progression free survival. These results suggest that coupling genetic immunoediting with activation of adaptive immunity is a relevant requisite for achieving long term clinical benefit by epigenetic immunomodulation in advanced melanoma patients.

Immune checkpoint inhibitors (ICI) are drugs targeting regulatory pathways in T cells to enhance antitumor immune responses[1]. Treatment with ICI has dramatically improved the clinical outcome of patients with tumors of different histotypes[2], including melanoma[3], and lung cancer[4]. However, the percentage of subjects who benefit from ICI therapy is still low, and novel therapeutic strategies are eagerly awaited to fully exploit their clinical potential. Indeed, even in the most responsive tumor types, both intrinsic[5] and acquired resistance[6,7] limit the efficacy of ICI therapy. The cellular and molecular characterization of human tumor samples by high-throughput and deep phenotyping approaches define the role of the immune microenvironment in driving the

✉ e-mail: maio@unisi.it

prognosis of cancer patients and their responsiveness to ICI therapies[8,9].

In this scenario, an active area of biomedical research aims to identify combinatorial approaches that could improve even the early phases of developing the antitumor response. Mechanistically, these new immunotherapy regimens aim at achieving one or more of three main effects thought to be crucial for overcoming resistance to immune intervention: (i) fostering the cross-talk between innate and adaptive arms of the immune system, (ii) promoting the recruitment of functional T cells at the tumor site, and (iii) counteracting recruitment/ function of immunosuppressive cells.

Among promising agents that may play a role in ICI combinations there are the hypomethylating agent (HMA) due to their immunomodulatory activity on tumor cells[10], the ability to activate innate immunity pathways[11,12] and the pre-clinical evidence for enhanced antitumor effects when combined with ICI[13]. In this scenario, our Italian Network for Tumor Biotherapy (NIBIT) Foundation Phase Ib NIBIT-M4 trial, based on the association of ipilimumab with the HMA guadecitabine in advanced melanoma patients, showed significant tumor immunomodulatory effects and preliminary evidence of promising clinical activity[11]. More recently, by comparing transcriptional programs elicited by different classes of epigenetic drugs in melanoma cells, we found that the main biological activity of guadecitabine is the promotion of gene expression and activation of master factors belonging to innate immunity pathways, including Type I–III interferon (IFN), NF-kB, and TLR[10]. These results corroborated the notion that the rescue of adaptive immunity by ICI may cooperate with the promotion of innate immunity by the HMA guadecitabine, thus potentially explaining the clinical activity of the combination. Two additional recent trials, combining guadecitabine with pembrolizumab in solid tumors[14,15] have indeed confirmed the significant antitumor activity of this combination in terms of clinical benefit rate (31.4%) or of progression-free survival (PFS) rate >24 weeks (37%). Crucially, in both studies, relevant immune effects were described regarding the upregulation of innate and adaptive immunity pathways in post-treatment samples.

Despite these initial promising clinical applications of the HMA guadecitabine combined with ICI, the further development of these combinatorial approaches requires the understanding of the biological mechanisms underlying the response and resistance to this specific type of epigenetic immunomodulation. The NIBIT-M4 study was a single-arm trial, therefore, in principle, such design prevents the possibility to achieve a sound disentanglement of the contribution of ipilimumab vs. the contribution of guadecitabine to the overall clinical activity.

Here we show that an advanced integrative systems biology approach, focusing on genomic, transcriptional, and methylation landscape analysis of baseline and on-treatment tumor tissues can shed light on the mechanisms of action of the two agents and on the resistance mechanisms likely impacting the ICI vs. the demethylating agent. In fact, first, we asked whether responder patients show differential enrichment of genomic features vs. non-responder patients. In particular, we analyze the effects of guadecitabine in the cases harboring mutations in its target genes. We also explored the promotion of expression of retroviral sequences and of repetitive elements that represent the mechanistic signature of immunomodulation by guadecitabine. We tested a combinatorial index for predicting long-term clinical benefit by integrating transcriptional information on the development of adaptive immunity, the Immunological Constant of Rejection (ICR)[16–18] with a measure of genetic immunoediting (GIE). The ICR signature incorporates IFN-stimulated genes driven by transcription factors IRF1 and STAT1, with CCR5 and CXCR3 ligands, immune effector molecules, and counter-activated immune regulatory genes. A high expression of ICR genes typifies 'hot'/immune active tumors characterized by the presence of a T helper 1 (Th1)/cytotoxic immune response and predicts survival and response to ICI in different

tumors including colon cancer[17], breast[19], bladder, stomach, head and neck[16], sarcoma[20], and melanoma[8]. The GIE index quantifies the amount of genetic immunoediting as the ratio between observed and expected tumor neoantigens. Here we show that stratification of NIBIT-M4 patients based on the ICR/GIE classification predicts Overall Survival (OS) and PFS, a finding that was validated in external larger ICI datasets. Collectively, our results contribute to improve the understanding of response and resistance to epigenetic immunomodulation, and provide a valuable multi-omics-related tool that may be used for patient stratification and clinical outcome prediction in immunotherapy cohorts.

## Results

### Long-term outcomes in the NIBIT-M4 trial

At data cutoff, July 1st 2022, with a minimum follow-up of 45 months, 6 (31%) of the 19 patients enrolled in the NIBIT-M4 study were alive. The median OS was 25.6 months (95% CI, 0.0-52.9), while the median PFS was 5.2 months (95% CI, 4.0–6.4); the 5-year OS rate was 28.9% with a 5-year PFS rate of 5.3%; median Duration of Response (DoR) was 20.6 months (95% CI, 12.4–28.8). Eighteen patients (95%) were treatment-naive at study entry and 1 (5%) had received PD-1 mAb as first-line therapy. Three patients were in Complete Response (CR) and off-study therapy, while 13/19 (68%) had received subsequent line(s) of therapy, including immunotherapy, target therapy, and/or chemotherapy; among those, 5 patients who had achieved a disease control (DC) had a median time to the subsequent treatment of 18.9 months (range 10.3–39.0) (Fig. 1).

### Genomic landscape: mutational profile differences in baseline and on-treatment lesions in responder (R) vs. non-responder (NR) patients

Longitudinal multi-omics profiling, including whole exome sequencing (WES), RNA Sequencing (RNASeq), and Reduced representation bisulfite sequencing (RRBS), were performed on tumor biopsies collected at baseline (week 0) and week 4 and week 12 on therapy from 14 patients (Supplementary Fig 1). Matched normal tissue collected at baseline was available for 8 patients. The exome sequencing profiling of our cohort, performed using stringent filtering, showed a high consistency of the somatic calls/mutations during treatment (Fig. 2, Supplementary Data 1). Although tumor mutational burden (TMB) was not significantly different in R vs. NR patients (16.9 vs. 15.3, $p$ value = 0.6, Student's $t$ test), several significant differences were found at the single gene level. BRAF was slightly enriched in NR ($p$ value = 0.02, Chi-squared test). In contrast, NRAS mutation was significantly more frequent in R vs. NR (50% vs. 0%, $p$ value = $5.4e^{-5}$, Chi-squared test). ADAMDEC1, encoding a disintegrin metalloproteinase associated with dendritic cell (DC) function, was altered only in lesions from R patients. Mutations in genes belonging to the epithelial to mesenchymal transition (EMT) pathway were enriched in NR ($p$ value = 0.01, Chi-squared test). The CDKN2A gene, frequently mutated in melanoma (35% of patients in TCGA cohort), was more frequently altered in NR patients. CDKN2A has been demonstrated to inhibit EMT and promote cancer immunity and CDKN2A deletions have been associated with ETM in different cancer types[21]. Three neuronal-related genes (PCLO, PLXNA4, and EPHA7), and the gene encoding the leptin receptor (LEPR), all reported as mutated in melanoma at a variable frequency (37%, 11%, 16% and 8% of samples in TCGA cohort, respectively), were altered more frequently in R compared to NR patients. The male germline-specific gene PLCZ1 was mutated only in lesions from R patients. Interestingly, several of the mutated genes (BRAF, NRAS, CDKN2A, EPHA7, PLXNA4) have been previously associated with response or resistance to ICI in monotherapy[22–26] although for some of them (e.g. BRAF) evidence is not conclusive[27]. The DNMT1 gene, encoding one of the guadecitabine targets, was mutated in two NR patients, and one of the mutations was a truncating event suggesting loss of function of the

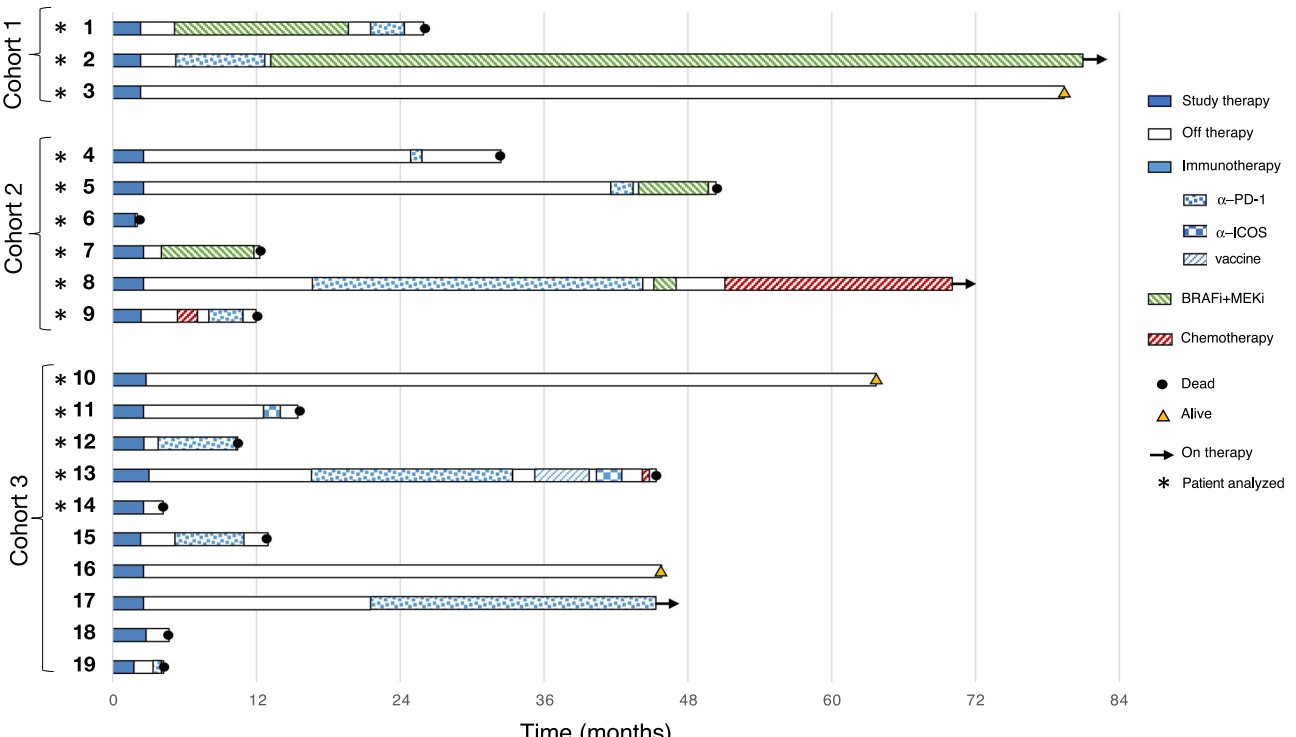

**Fig. 1 | Swimmer plot analysis of NIBIT-M4 patients.** Swimmer plot showing by study arm patients who at the time of data cutoff were alive and either still on study treatment or off-study therapy, without having received subsequent therapy, and all patients who have received subsequent treatment at the time of data cutoff, regardless of whether alive or dead. Subsequent treatments include immunotherapy (i.e., anti-PD-1 monotherapy or combinations, ICOS agonist or vaccine), BRAFi + MEKi, and chemotherapy.

*DNMT1* gene product. An additional mutation of *SETD2*, involved in chromatin organization, was observed in another NR patient. We analyzed the effect of somatic mutations in *DNMT1* or *SETD2* on the methylation during therapy. The increasing or decreasing trend was evaluated based on the slope of the robust linear regression line between the three time points. We analyzed all genomic regions, specifically the coding, intergenic, intronic regions (Supplementary Fig 2a), and then regulatory regions (Supplementary Fig 2b) in *DNMT1*- and *SETD2*-mutant vs. wild-type lesions. Interestingly, *DNMT1*- or *SETD2*-mutant samples did not show the decreasing pattern over time that we observed in the wild-type lesions. This was also confirmed in long terminal repeat (LTR) including endogenous retroviral elements (ERVs) (Supplementary Fig 2c). ERVs are particularly important for the immune response to tumors, even in the context of ICI[28]. Previous studies have shown that by inducing the re-expression of ERV sequences, demethylating agents can activate the viral mimicry response secondary to intracellular recognition of viral dsRNA[29,30]. This response explains the promotion of type I IFN and innate immunity pathways that characterize the guadecitabine-specific gene signature recently defined by us in melanoma cells[10]. We confirmed that the expression of LTR elements was inversely correlated with the expression after treatment with demethylating agent in wild-type, but not in mutant tumors (Supplementary Fig 2d).

Solar ultraviolet (UV) radiation is one the main etiological factor for skin cancer, including melanoma as it causes a characteristic genomic mutational pattern associated with elevated TMB via the formation of pyrimidine-pyrimidine photodimers (COSMIC signature 7)[31,32]. We performed mutational signature deconvolution on the cases for which the matched normal was available ($n = 8$). Two (SBS7a and SBS7b) of the four UV-associated mutational signatures were the most frequently observed (Supplementary Fig 3). However, the limited number of cases did not allow for detecting any significant association between UV mutational signature rate and response ($p$ value = 0.064,

Chi-squared test), mutation load ($p$ value = 0.435, Chi-squared test) and neoantigen load ($p$ value = 0.122, Chi-squared test), respectively.

Overall, our longitudinal analysis confirmed several genomic features previously associated with response to ICI, such as defects in the EMT, mutations of *BRAF* and *NRAS* and in other immune-related genes, and was useful to discover, even for a limited number of cases, that loss-of-function mutations in chromatin organization and guadecitabine targets may contribute to limit the efficacy of the combined therapy and the epigenetic immune-modulatory effect of the HMA.

### Transcriptional landscape of baseline and on-treatment tumor lesions: distinct and evolving transcriptional programs distinguish R from NR patients

RNA-sequencing data from the NIBIT-M4 were used to carry out differential gene expression analysis between R and NR patients at different time points of treatment (Supplementary Data 2). This analysis showed a progressive enrichment from baseline to week 12 in Gene Ontology Biological Processes (GO: BP) categories related to immune processes in R compared to NR patients (Fig. 3a and Supplementary Data 3). In contrast, in lesions from NR patients, a progressive increase from baseline to week 12 was found for GO terms related to adhesion, cell cycle, metabolism, and skin developmental processes. We tested several state-of-the-art predictive signatures of response to ICI, including MIRACLE score[8], ICR[19] IMPRES[33], TIDE[34], and MPS[35]. None of these five scores discriminated against R from NR patients when considering either baseline and on-treatment samples or at various time points (Supplementary Fig 4a and Supplementary Data 4) neither using standard tests for comparing the difference between multiple subjects and repeated measures.

We have recently performed a comparative profiling of gene signatures induced by different classes of epigenetic drugs in melanoma cell lines[10] and found that guadecitabine activates several innate immunity pathways by induction of a signature of 166 genes. To better

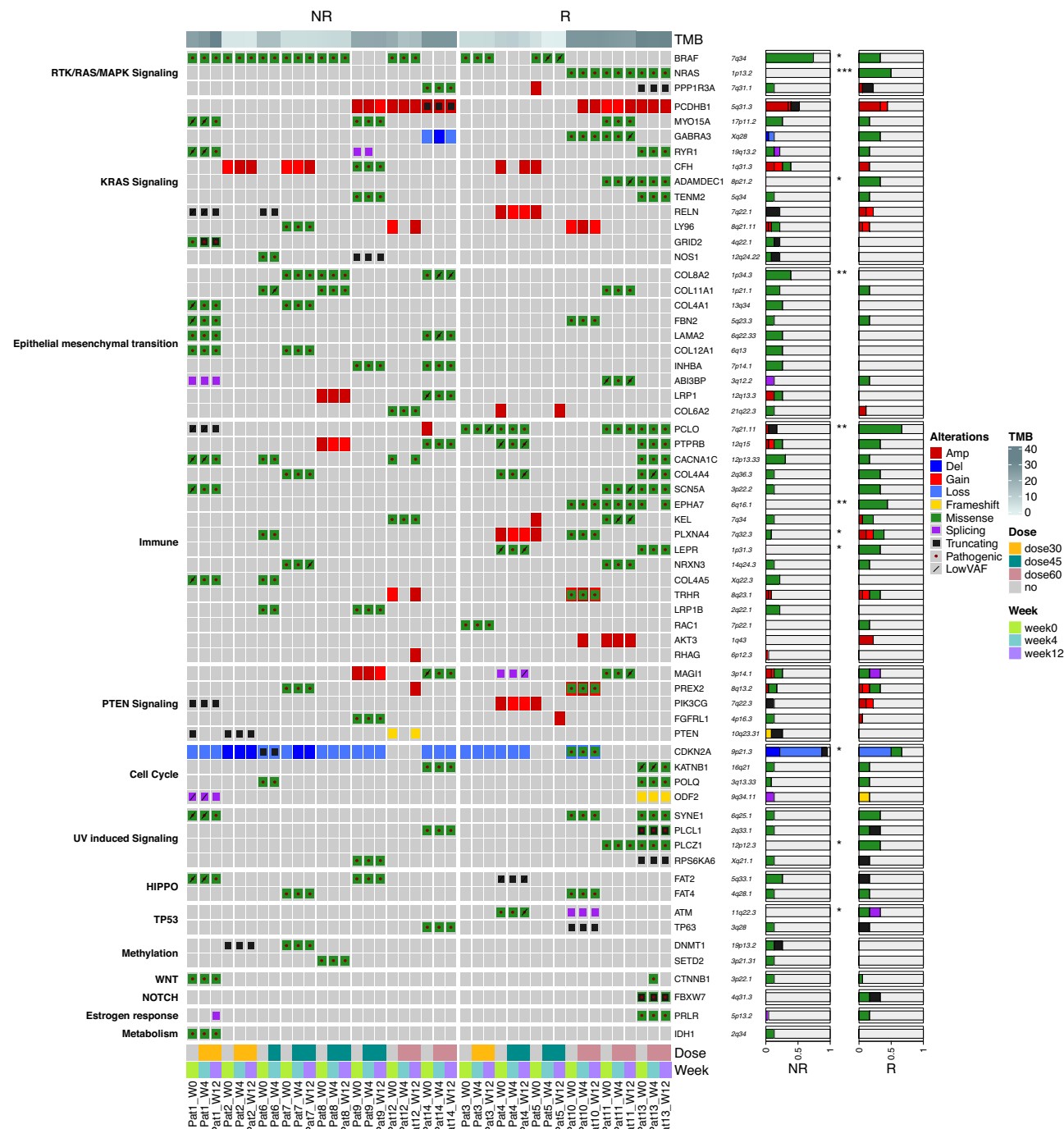

**Fig. 2 | Genomic landscape of the NIBIT-M4 trial.** Oncoplot of frequent somatic nonsynonymous and copy number alterations of NIBIT-M4 trial organized by response (columns, non-responder (NR, $n = 8$) and responder (R, $n = 6$) patients) and pathways (rows). Tumor Mutation Burden (TMB), dose (30, 45, and 60 mg/m²/ day), and time (week 0, week 4, week 12) of treatments are indicated. The proportion of alterations in NR and R groups is visualized for each gene ($p$ value of two-sided Pearson's chi-squared test statistic, *$p < 0.05$, **$p < 0.01$, ***$p < 0.001$). Source data are available in Supplementary Data 1.

disentangle the effect of the HMA in our cohort, we performed gene set enrichment analysis of the guadecitabine-specific signature on the list of differentially expressed genes between R and NR (Fig. 3b). We found that this signature was significantly activated in R at week 4 ($p$ value = 4.172e$^{-05}$, GSEA permutation test) and week 12 ($p$ value = 1.055e$^{-04}$, GSEA permutation test). To further validate this finding, we used the data from another trial in which patients with ovarian cancer were treated with a combination of ICI and epigenetic drug[14]. We observed a significant activation of the guadecitabine signature after treatment in R patients (Fig. 3c). Our results confirm the main immune-modulatory effect of HMA and that this effect is

associated with response. By a custom-designed NanoString assay we then explored differential expression in R vs. NR lesions of 20 published immune-related signatures (Supplementary Fig 4b) providing information on B-cell content and differentiation[34,36], tertiary lymphoid structures (TLS) formation[37,38], follicular T helper (TFH) cells[34,36], T-cell exhaustion (TEX) subsets[39], tumor-associated endothelial cells[40], immune checkpoint blockade (ICB) response[41,42], and the recently identified guadecitabine-specific signature genes induced by this demethylating agent in melanoma cell lines[10]. The large majority of these signatures was selectively enriched, considering all time points, in tumor biopsies from R compared to NR patients. These results were

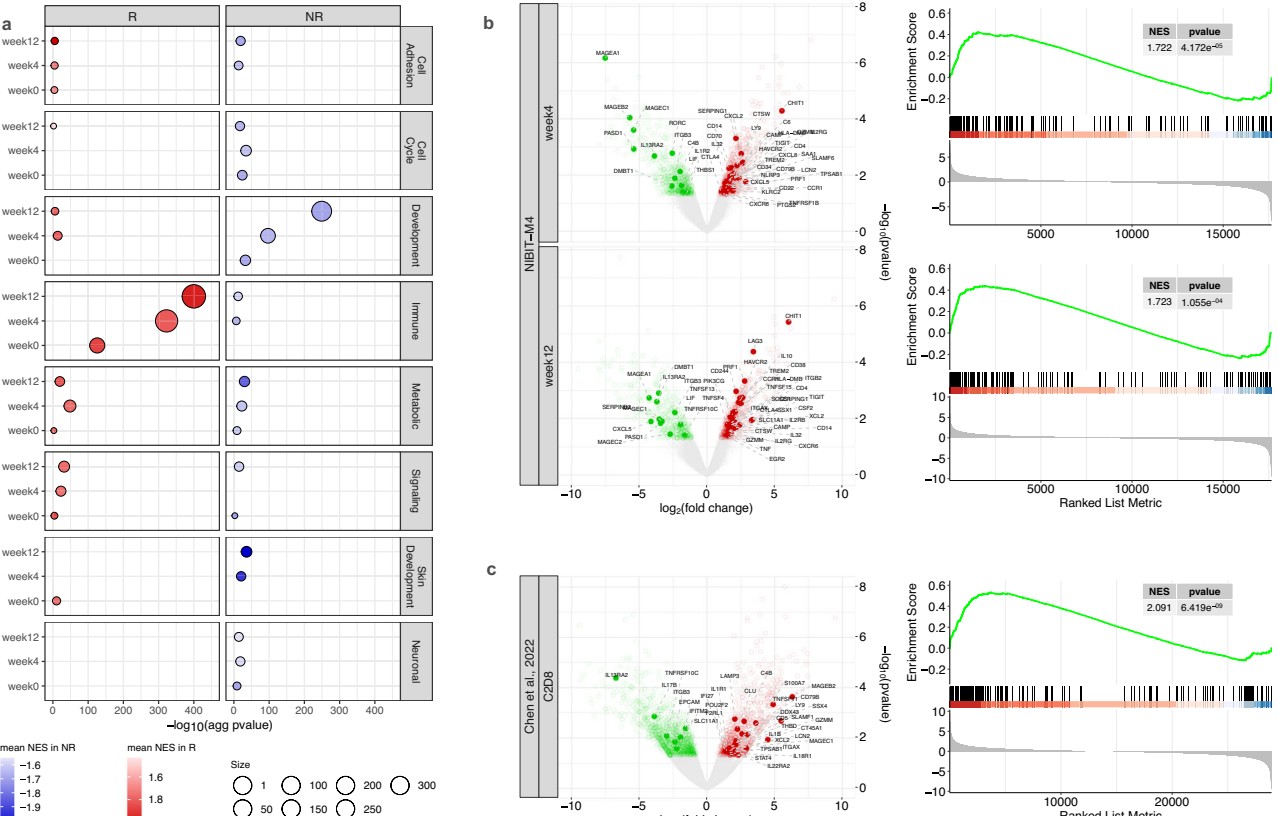

**Fig. 3 | Transcriptional landscape of the NIBIT-M4 trial. a** Gene Set Enrichment Analysis (GSEA, as implemented in clusterProfiler[84]) from supervised differential analysis between R ($n = 6$) and NR ($n = 8$) patients (negative binomial generalized linear model with likelihood ratio test (glmLRT), as implemented in EdgeR[82]) before treatment (week 0) and after four (week 4) and twelve (week 12) weeks. $x$ axis reports the aggregated $p$ value of significant enriched GO:BP terms (false discovery rate method from empirical permutation test, FDR < 0.1), computed using Fisher method and classified into seven main categories. Size of the dot represents the number of GO:BP terms grouped into a category; color of the dots represents the mean Normalized Enrichment Score (NES) of the terms. **b** Volcano plot (left) of differentially expressed genes ($p$ value < 0.05 from glmLRT, as implemented in EdgeR[82]) between R ($n = 6$) and NR ($n = 8$) patients, labeled genes belong to the guadecitabine-specific gene signature[10]. $x$ axis reports the effect size (in log scale), $y$ axis reports the $-\log(p$ value) from glmLRT, as implemented in EdgeR[82]. GSEA enrichment (right) of the guadecitabine-specific signature on the ranked list of differentially expressed gene at week 4 and week 12. **c** Same as in **b** using the dataset from a trial of combined therapy ICI plus HMA from Chen et al. 2022[14] (C2D8 post-treatment, $n = 4$ R and $n = 5$ NR). Source data are provided as a Source Data file.

consistent with preferential development in R lesions of a coordinated T- and B-cell mediated immune response involving TLS and TFH cells, with enhanced expression of IFN-γ-induced genes crucial for ICB response and with increased presence of CD8⁺ T cells at different stages of exhaustion.

We also used Ingenuity Pathway Analysis (IPA) to identify canonical pathways differentially regulated in R vs. NR patients during treatment. Indeed, IPA database accounts for positive and negative associations of genes and pathways. This analysis predicted significant activation in R patients of several immune-related pathways including the "Pathogen induced cytokine storm", the "TH1" and "TH2", the "phagosome formation" and the "cross-talk between DC and Natural Killer (NK) cells" pathways (Supplementary Fig 4c). In contrast, pathways predicted to be inhibited in R vs. NR patients included the "PD-1, PD-L1 cancer immunotherapy", the "MSP-RON signaling in macrophages" and the "GP6 signaling" pathways. These results confirm that R patients experience the activation of immune pathways crucial for developing innate and adaptive immunity.

Finally, we analyzed the differential expression between R and NR patients in selected gene sets (Supplementary Fig 5). Lesions from R patients showed a progressive increase of expression, from baseline to week 12, of genes encoding for molecules controlling T-cell activation, inhibitory receptors and ligands, chemokines, and components of the immunoproteasome. In line with our previous reports where we have

shown that DNA methyltransferase inhibitors can upregulate the expression of major histocompatibility complexes (MHC) proteins[43], we observed that HLA class I and class II were significantly upregulated in R patients.

By contrast, lesions from NR patients showed higher expression of cell cycle-, EMT- and skin development-related genes with a consistent pattern across the weeks. Several of the EMT-related genes with higher expression in NR patients encoded for molecules controlling adhesion (*ITGB3*, *VCAM1*, collagens, and others), interaction with extracellular matrix (*VEGFA*, *MMP14*, *WNT5A*, *LAMA1*, and others), and melanoma dedifferentiation (*KRT9*, *KRT10*, *EGFR*, and others). In particular, *WNT5A* is a well-defined feature of a poor melanoma phenotype and has been associated with a negative modulation of the tumor microenvironment[44]. Lesions from NR patients also showed significantly higher expression, compared to R patients, in several key genes controlling cell proliferation including cyclin-dependent kinases 1 and 4, cyclins B1 and B2, mitotic checkpoint serine/threonine kinase B, and other transcription factors controlling cell cycle such as *E2F1* and *E2F2*.

The composition of the tumor microenvironment of our cohort was deconvolved from transcriptome profiling data using eight immune- and two stromal-cell signatures with MCP-counter[45] (Fig. 4a). We found a significantly higher abundance in R vs. NR of CD8⁺ T cells ($p$ value = 0.042, Wilcoxon test) at week 4 (Fig. 4a), other comparisons

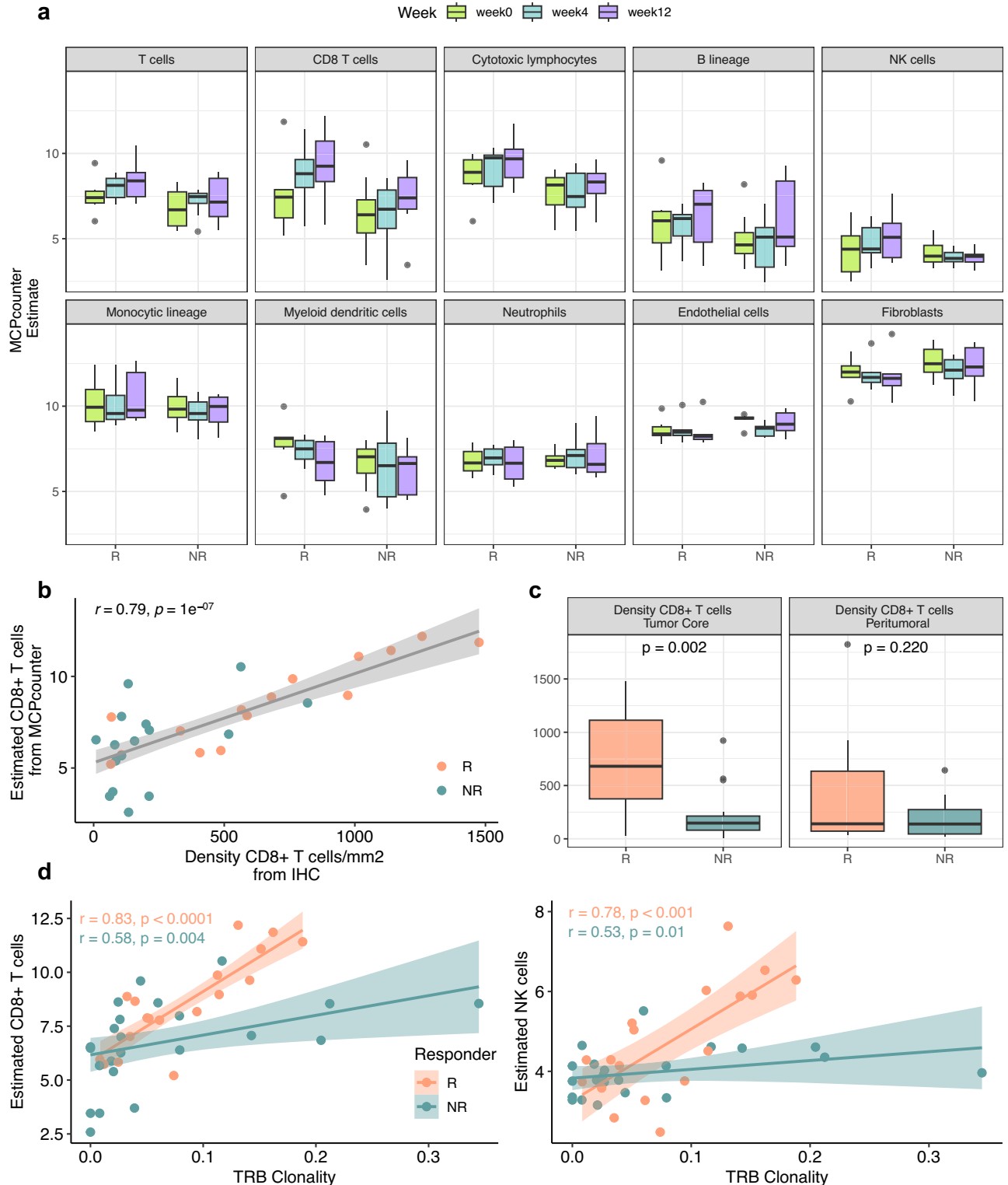

**Fig. 4 | Immune microenvironment. a** Immune microenvironment deconvolution of immune cell fractions stratified by time point and response ($n = 6$ R and $n = 8$ NR). **b** IHC validation (x axis) of the deconvolution of CD8[+] T-cell proportion estimated from RNASeq (y axis) (Spearman correlation coefficients rho ($r$) and associated $p$ values from two-tailed correlation test are provided for R and NR groups, samples from $n = 15$ R and $n = 16$ NR). **c** Density of CD8 T cells by location from IHC in the tumor core (samples from $n = 15$ R and $n = 16$ NR), and peritumoral (samples from $n = 9$ R and $n = 13$ NR) ($p$ value from two-sided Student's $t$ test between R and NR groups). **d** Scatterplot between the T-cell receptor clonality (B locus) and CD8[+] T-cell (left) and NK cell abundances (right) (Pearson's correlation coefficient ($r$)) and associated $p$ values from two-tailed correlation test are provided for R ($n = 6$) and NR ($n = 8$) groups. **a, c** Box plots show the median as center, the lower and upper hinges that correspond to the 25th and the 75th percentile, and whiskers that extend to the smallest and largest value no >1.5*IQR. Values that stray more than 1.5*IQR upwards or downwards from the whiskers are considered potential outliers and represented with dots. **b, d** Bands represent confidence intervals ($\pm 0.95$) around a linear model fitted by robust regression using an M estimator. Source data are provided as a Source Data file.

did not produce significant differences also due to the low number of cases. The estimated increased abundance of CD8[+] T cells in R lesions was in agreement with available quantitative immunohistochemistry (IHC) data ($r = 0.79$, $p$ value $= 1e^{-07}$, N = 11, Spearman correlation test) for this immune subset (Fig. 4b) and with enhanced CD8[+] intratumoral T cells in R lesions ($p$ value $= 0.002$, Student's t-Test) (Fig. 4c).

We then used the clonality of V(D)J rearrangements within the TCR Beta locus (TRB) as a proxy for estimating T-cell expansion during therapy. TRB clonality was significantly correlated with the estimated abundances of CD8[+] T cells ($r = 0.83$, $p$ value $< 0.0001$, Spearman correlation test) and NK ($r = 0.78$, $p$ value $< 0.001$, Spearman correlation test) cells in R rather than NR patients (Fig. 4d).

Taken together, the gene expression landscape of NIBIT-M4 lesions indicated that distinct and evolving transcriptional profiles characterized baseline and on-treatment tumor biopsies from R compared to NR patients. Lesions from R patients showed progressive enrichment for signatures and gene sets revealing activation of adaptive immunity and effective immunomodulation by guadecitabine with a preferential and clonal activation of T-cell subpopulation and NK cell in the tumor microenvironment. Lesions from NR patients revealed lack of promotion of immunity in a tumor transcriptional background dominated by proliferation and EMT processes.

## Integrative analysis of methylation and transcriptomic profiles during treatment

The availability of the longitudinal sampling of tumor biopsies gave us the opportunity to evaluate the immune-modulatory effect of the HMA during therapy. Although NIBIT-M4 trial was not designed to compare the effect of the combination HMA plus ICI vs. ICI alone, we could evaluate the changes of the methylation pattern induced by the treatment with the HMA. When we considered the overall methylation level across the genome, a trend towards global demethylation was observed for both R and NR patients (Supplementary Fig 6a) as well as in specific genomic regions, the effect of the HMA in the coding part of the genome (exons and intergenic regions) had a decreasing trend in R and NR patients. Similarly, for promoters, UTRs regions, and other regulatory regions such as enhancers and super-enhancers (Supplementary Fig 6a) NR had a lower decreasing trend than R patients. We evaluated the specific methylation pattern during therapy in Long interspersed nuclear elements (LINEs), Short interspersed nuclear elements (SINEs), and LTR. The effect of HMA was a general decreasing trend of methylation for both R and NR patients (Fig. 5a). The relationship of this demethylation process with the changes in expression of these repeated element of the genome was also considered. We correlated the expression and methylation of different

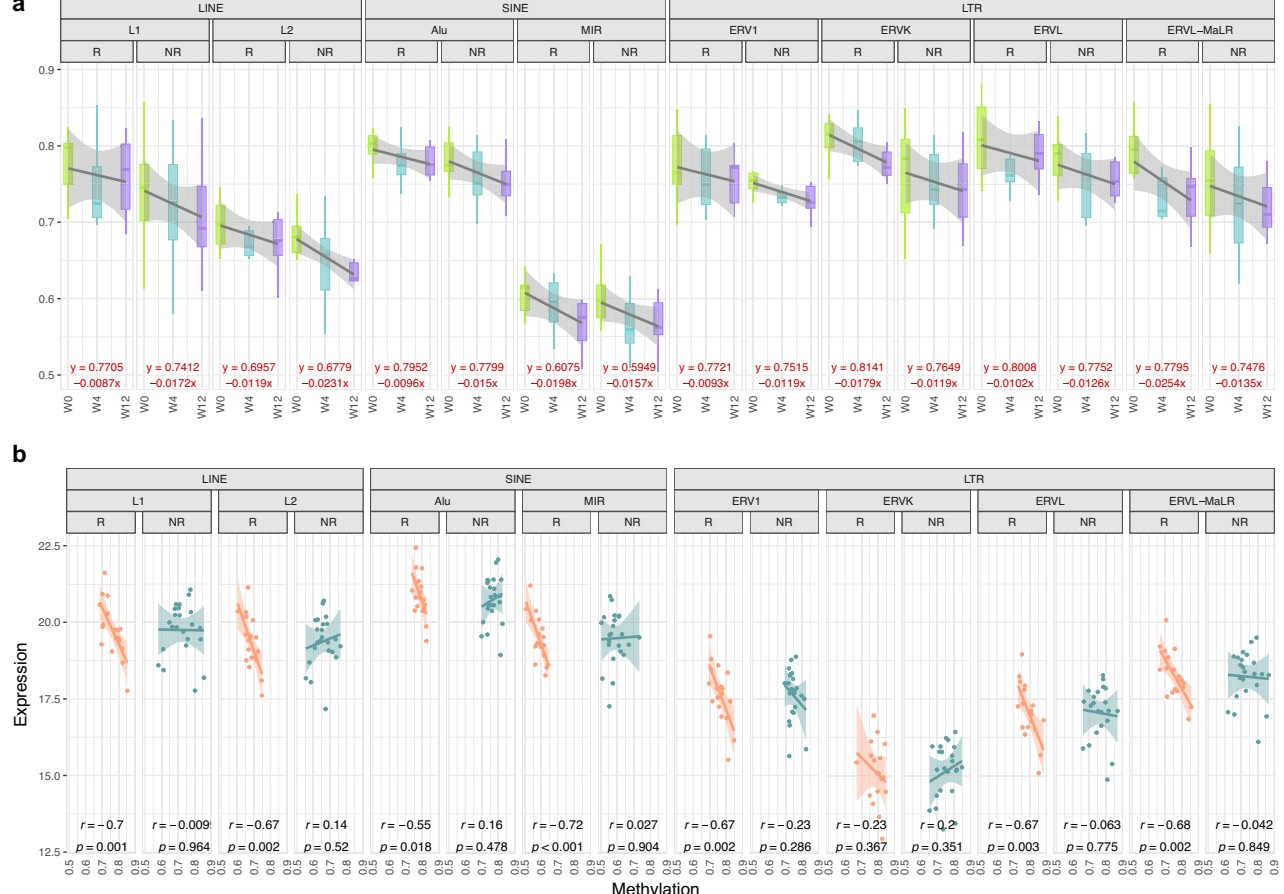

**Fig. 5 | Evolution of the overall methylation pattern in various genomic regions and regulatory regions as function of time between R and NR patients.** **a** Variation of the overall methylation pattern in Long interspersed nuclear elements (LINEs), Short interspersed nuclear elements (SINEs) and Long terminal repeat (LTR) endogenous retroviral elements (ERV). The increasing or decreasing trend was evaluated based on the inclination of the robust linear regression line between the three time points. Plots report the values at different time points divided between R ($n = 6$) and NR ($n = 8$). Box plots show the median as center, the lower and upper hinges that correspond to the 25th and the 75th percentile, and whiskers that extend to the smallest and largest value no more than 1,5*IQR. Values that stray more than 1.5*IQR upwards or downwards from the whiskers are considered potential outliers and represented with dots. **b** Scatterplot between methylation and expression in R and NR patients in SINE, LINE and LTR elements (Pearson's correlation coefficient ($r$)) and associated $p$ values from two-tailed correlation test are provided for R ($n = 6$) and NR ($n = 8$) groups. **a, b** Bands represent confidence intervals ($\pm0.95$) around a linear model fitted by robust regression using an M estimator. Source data are provided as a Source Data file.

LINE, SINE, and LTR elements including ERVs. The number of reads that mapped on these elements was used as a proxy of their expression (see Methods). This analysis showed a significant anti-correlation between methylation and expression only for R patients, whereas for NR patients a similar inverse trend was not observed (Fig. 5b).

Finally, we performed differential promoter methylation and gene expression analysis between R and NR patients (Supplementary Fig 6b). We used SMITE[46] to rank the upregulated and hypomethylated genes as well as the downregulated and hypermethylated genes between R and NR patients at different time points. Consistently with the previous observations, functional enrichment of hypomethylated and upregulated genes between R and NR patients resulted in biological processes associated with immune functions at week 12. Genes enriching immune response included granzymes (*GZMM*), interleukins and interleukin receptors (*IL12, IL32, IL15, IL12RB1*), cytokines and cytokine receptors (*TNF, CXCR4, TNFRSF1B*) and HLA genes and trans-activators (*CIITA, HLA-DOA, HLA-DMB, HLA-E*).

We also interrogated the effect of the epigenetic agent in the set of lesions from NR patients where we did not find evidence of promotion of immunity as in the lesions from R patients. To answer this question, we jointly compared the expression and methylation profiles of NR patients at different time points. We applied SMITE to the differential expression and methylation analysis between week 12 and the baseline in NR patients. This analysis aimed at uncovering the pathways that were activated on-treatment in NR patients. The top 100 genes of the ranked list were used for functional enrichment. Some of these genes were significantly associated with fate determination (*WNT5A*), cell migration (*PRKD2*), epithelial differentiation (*KRT15*), while most frequent GO:BP terms included pathways related to "development", "differentiation" and "migration/motility" (Supplementary Fig 7a). On the contrary, after treatment, the same longitudinal analysis of lesions from R patients resulted in the activation of immune response functions such as regulation of T and NK cell activation and proliferation (*IL15*), Th1 polarized T cells (*TBX21*), and other functional categories associated with lymphocyte activation (Supplementary Fig 7b).

Overall, our results showed that treatment with HMA induces the demethylation of genomic regions, in particular transposable elements, and ERVs. The effect of these epigenetic changes was associated with their higher expression in R patients. Moreover, intersection of differentially expressed and methylated genes showed specific epigenetic activation of immune functions in R patients.

## The ICR/GIE classification contributes to explain response, resistance through immune escape and long-term clinical outcome

TMB and neoantigen loads were highly correlated ($r = 0.77$, $p$ value $5e^{-09}$, Spearman correlation test) in the lesions of the NIBIT-M4 trial, however none of these two parameters discriminated against R from NR patients.

According to the immunoediting theory, tumor clones can be eliminated by CD8+ T cells, resulting into a depletion of tumor-associated antigens, including neoantigens[47]. Immune-edited lesions are therefore expected to have less neoantigens then expected, while tumors that are not able to elicitate an antitumor immune response will not display sign of immunoediting.

We then tested the hypothesis that the genetic immunoediting (GIE) score[48,49], an index that integrates both TMB and neoantigen load information in a single measure of the extent of immunoediting, could show an association with clinical response.

The GIE value was calculated as the ratio between the observed vs. expected number of neoantigens in each tumor sample (Supplementary Data 5). The expected number of neoantigens was estimated by training a linear model having TMB as the independent variable and neoantigen load as the outcome variable. Tumors with a number of

neoantigens lower than expected (i.e., lower GIE values) are thought to display evidence of immunoediting, whereas a higher frequency of neoantigens than expected indicates a lack of immunoediting (Non-GIE). However, the difference was not significant at the baseline, possibly due to the low number of cases (Supplementary Fig 8a). We then implemented a more precise metric capturing the extent of immunoediting, by combing both information of GIE and the presence of a robust intratumoral cytotoxic immune response (ICR)[17]. We then speculated that the combined presence of an adaptive cytotoxic immune response (high ICR) and the evidence of neoantigen depletion (GIE), would more precisely capture "truly" immune-edited lesions. We then stratified tumor lesions from patients enrolled in the NIBIT-M4 study based on GIE score greater or lower than one (coded respectively as "Non-GIE" and "GIE") and ICR score greater or lower than zero, yielding four groups: High-ICR/GIE, High-ICR/Non-GIE, Low-ICR/GIE, Low-ICR/Non-GIE (Fig. 6a). The tumor samples belonging to R patients were highly enriched in the High-ICR/GIE group (61%, $p$ value = $4.2e^{-04}$, Chi-squared test). The ICR/GIE classification was significant even when limiting the analysis on baseline lesions (67%, $p$ value = $2.1e^{-02}$, Chi-squared test) with four of the five lesions in the High-ICR/GIE group (Supplementary Fig 8b). To shed light on the mechanism that differentiates GIE in the presence of adaptive immunity captured by ICR, we then performed a supervised transcriptome analysis comparing the High-ICR/GIE vs. the High-ICR/Non-GIE groups (Fig. 6b). This analysis showed that the High-ICR/GIE group (i.e., the group in the upper left quadrant, Fig. 6a) was characterized by enhanced representation of several Biological Processes related to immune response including antigen processing and differentiation (Fig. 6b). This also suggested that the "High-ICR/Non-GIE" group (upper right quadrant in Fig. 6a) could be defective for expression of antigen processing and presentation genes in a way that could explain the lack of immunoediting. In agreement, lesions from the High-ICR/Non-GIE group showed loss of expression of HLA class I antigens on tumor cells despite the presence of CD8+ T cells infiltrating the tumor core (Fig. 7 and Supplementary Fig 9a).

We then asked whether the patient groups defined by the ICR/GIE classification also experienced different long-term clinical outcomes. By stratifying patients according to the ICR/GIE classification of week 12 biopsies we found a slightly significant difference ($p$ value = 0.035, log-rank test) in OS and in PFS ($p$ value = 0.022, log-rank test) between the High-ICR/GIE and the High-ICR/Non-GIE group (Fig. 6c and Supplementary Fig 9b). The ICR/GIE stratification remained significant for OS and PFS even when taking into consideration all four subsets (Supplementary Fig 9b). In contrast, patients' classification by response groups was not associated with OS (Fig. 6c), although it was associated with PFS (Supplementary Fig 9b).

Comprehensive assessment of the ICR, GIE, CD8+ T cell and HLA Class I scores, available from lesions of 11 patients (Supplementary Fig 10a), indicated that the overall profile for these four parameters was consistent with the observed clinical response in all but one patient (#11). Patient #11 had a Low-ICR/Non-GIE score, but was classified as R patient due to stable disease (SD) in target lesions. Eventually, the patient underwent disease progression in target lesions and developed several new lesions (Supplementary Fig 10b).

Collectively, these findings provided a mechanistic explanation for the genesis of the High-ICR/Non-GIE subset.

## Validation of the ICR/GIE classifier in other response datasets
To validate the ICR/GIE stratification, we assembled a cohort of 83 melanoma cases treated with ICI, either anti-CTLA4 or -PD-1, from previous published studies[5,50,51] for which TMB, neoantigen load and gene expression were available. The stratification of patients into the 4 ICR/GIE subsets confirmed the enrichment of R patients in the High-ICR/GIE group (Fig. 8a). Significant differences in OS between patients characterized as High-ICR/GIE vs. those coded as High-ICR/Non-GIE

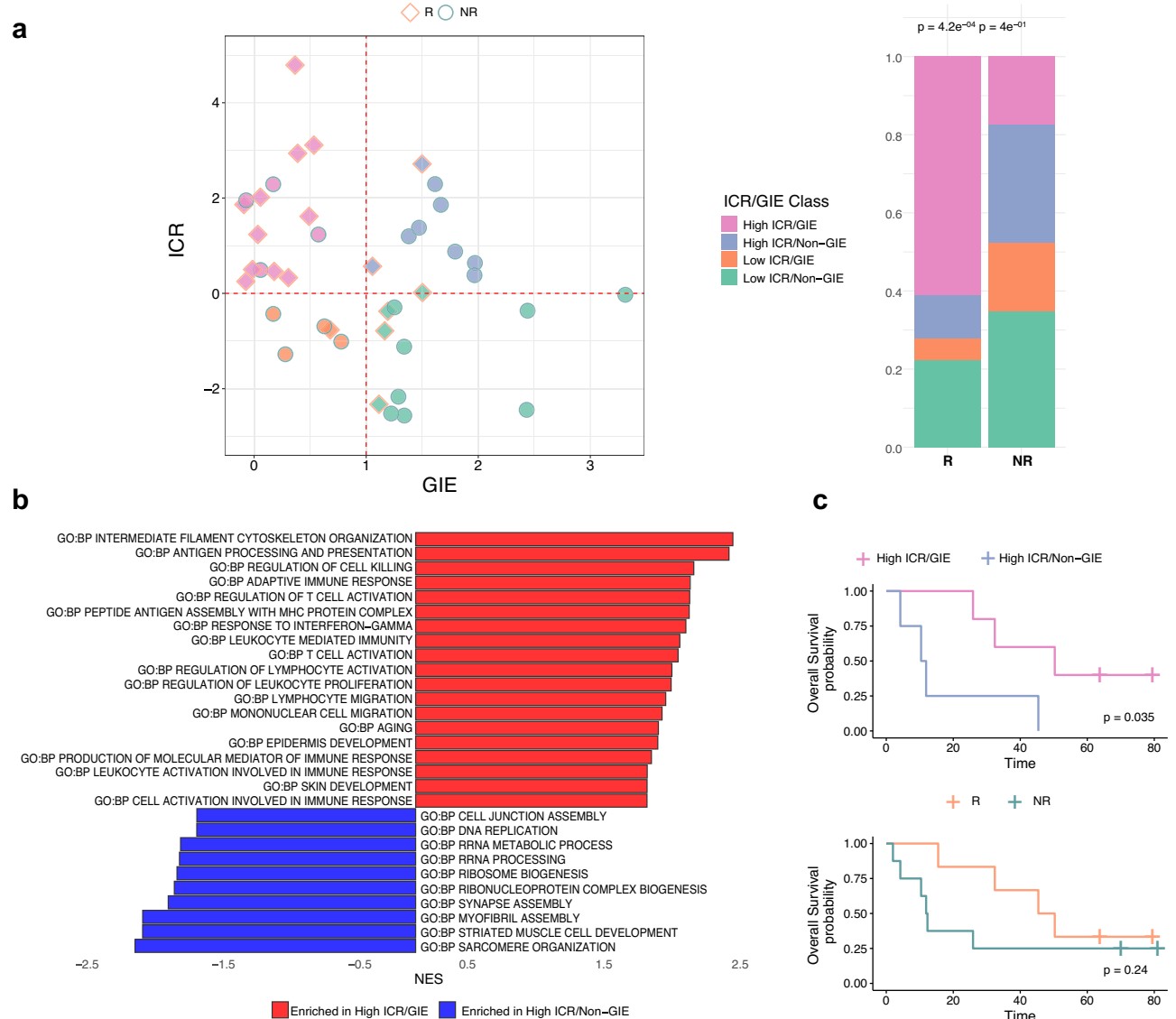

**Fig. 6 | Biomarkers of immunoediting (1). a** Scatterplot of ICR score by Genetic Immunoediting (GIE) score for R ($n = 18$) and NR ($n = 23$) samples (left) and their proportion (right) after classification as ICR/GIE classes ($p$ value from two-sided Pearson's chi-squared test statistic). **b** Barplot of most significantly (FDR < 0.01) enriched GO:BP terms from GSEA analysis of High-ICR/GIE ($n = 15$) vs. High-ICR/ Non-GIE ($n = 9$) samples' comparison. **c** Kaplan–Meier for OS by patients classified as High-ICR/GIE ($n = 5$) or High-ICR/Non-GIE ($n = 4$) at week 12 (top) and R ($n = 6$) or NR ($n = 8$) (bottom). Time is indicated in months and censor points are indicated by vertical lines. $p$ values are calculated by log-rank test. Source data are provided as a Source Data file.

were observed even in this validation cohort (Fig. 8b). Interestingly, differential pathway analysis in the High-ICR/GIE vs. High-ICR/Non-GIE subsets identified the Biological Process "antigen processing and differentiation" as selectively enriched in the High-ICR/GIE subsets as we found in the NIBIT-M4 cohort (Fig. 8c). These results suggest that the same mechanism uncovered in the NIBIT-M4 cohort could explain the High-ICR/Non-GIE subset even in the validation cohort: a defective antigen processing and presentation pathway may contribute to suppress genetic immunoediting even when lesions have a high ICR profile, and this may be a general phenomenon irrespective of the type of immunotherapy that is being used.

Collectively, these results suggested that effective coupling of tumor immunoediting with activation of adaptive immunity can promote response and improved clinical outcome in the NIBIT-M4 epigenetic immunomodulation trial. In contrast, defective development of adaptive immunity or lack of genetic immunoediting, associated with immune escape mechanisms, may favor resistance and less favorable long-term clinical outcome.

## Discussion

The NIBIT-M4 trial has been the first Phase Ib epigenetic immunomodulation study testing the association of the HMA guadecitabine with ICI in solid malignancies. Treatment of metastatic melanoma patients with guadecitabine combined with ipilimumab was found to be safe, feasible, and tolerable, with initial signs of clinical and immunologic activity[11]. The 5-year survival rate, duration of response, and time to subsequent treatment in patients who achieved a DC we report here are intriguing and clinically meaningful. These findings compare favorably with the efficacy of anti-CTLA-4 monotherapy in metastatic melanoma patients[52], though with the limitations of interstudy comparisons that need to be interpreted with caution and to be placed in context. Nevertheless, the long-term follow-up of the NIBIT-M4 trial seems to support the clinical potential of guadecitabine combined with anti-CTLA-4 therapy, though the relative contribution of each agent and the potential of guadecitabine maintenance therapy in the clinical results we observed could not be fully dissected in this trial. In this scenario, our ongoing randomized phase II NIBIT-ML1 study

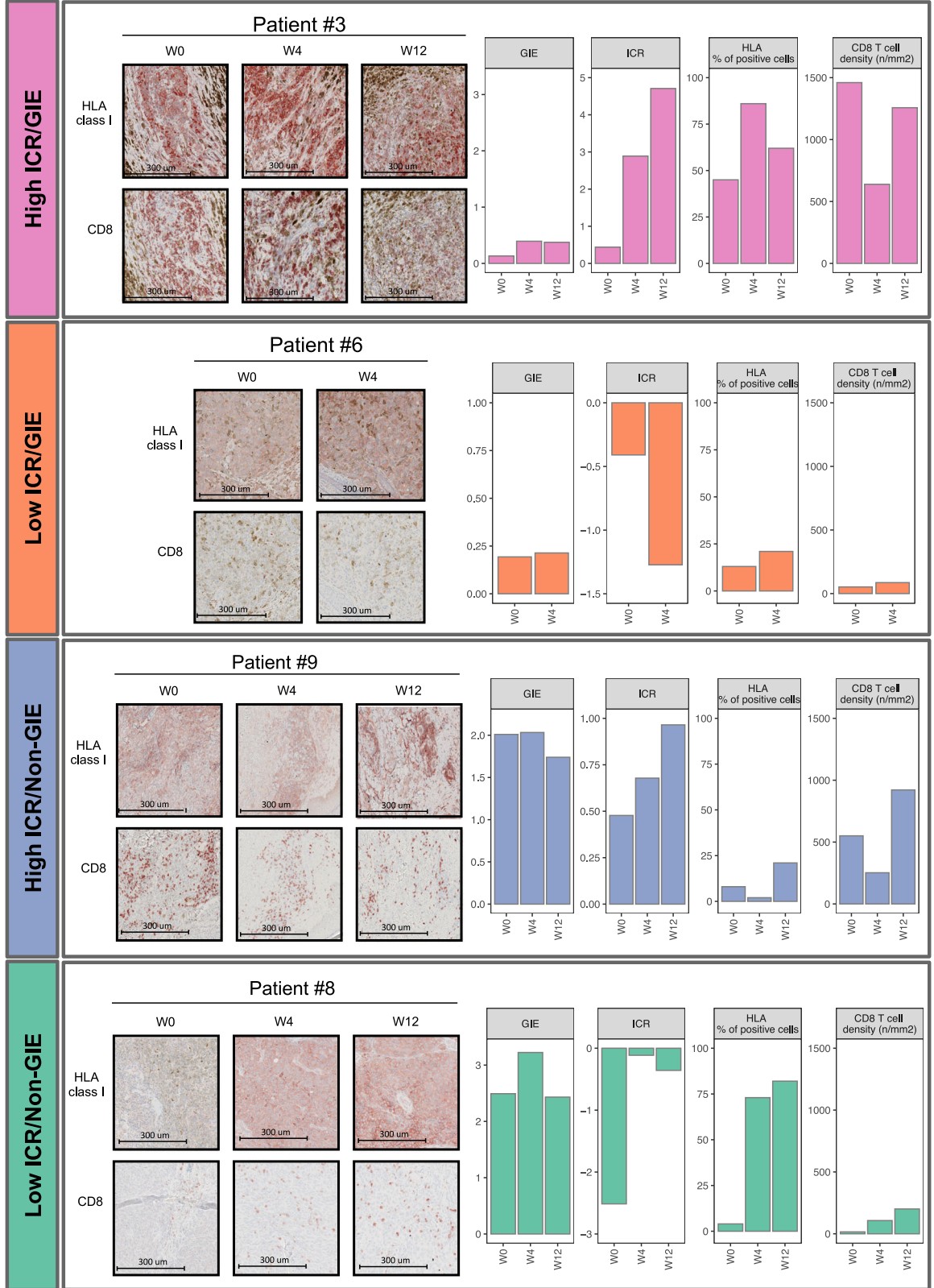

**Fig. 7 | Biomarkers of immunoediting (2).** Microphotographs of HLA class I and CD8 immunohistochemistry for representative patients in each ICR/GIE class (left). The plots represent the ICR and GIE sample scores, percentages of HLA-positive cells, and density of CD8 T cells. The presented data was derived from one experimental run independently reviewed by three different trained scientists. Source data are provided as a Source Data file.

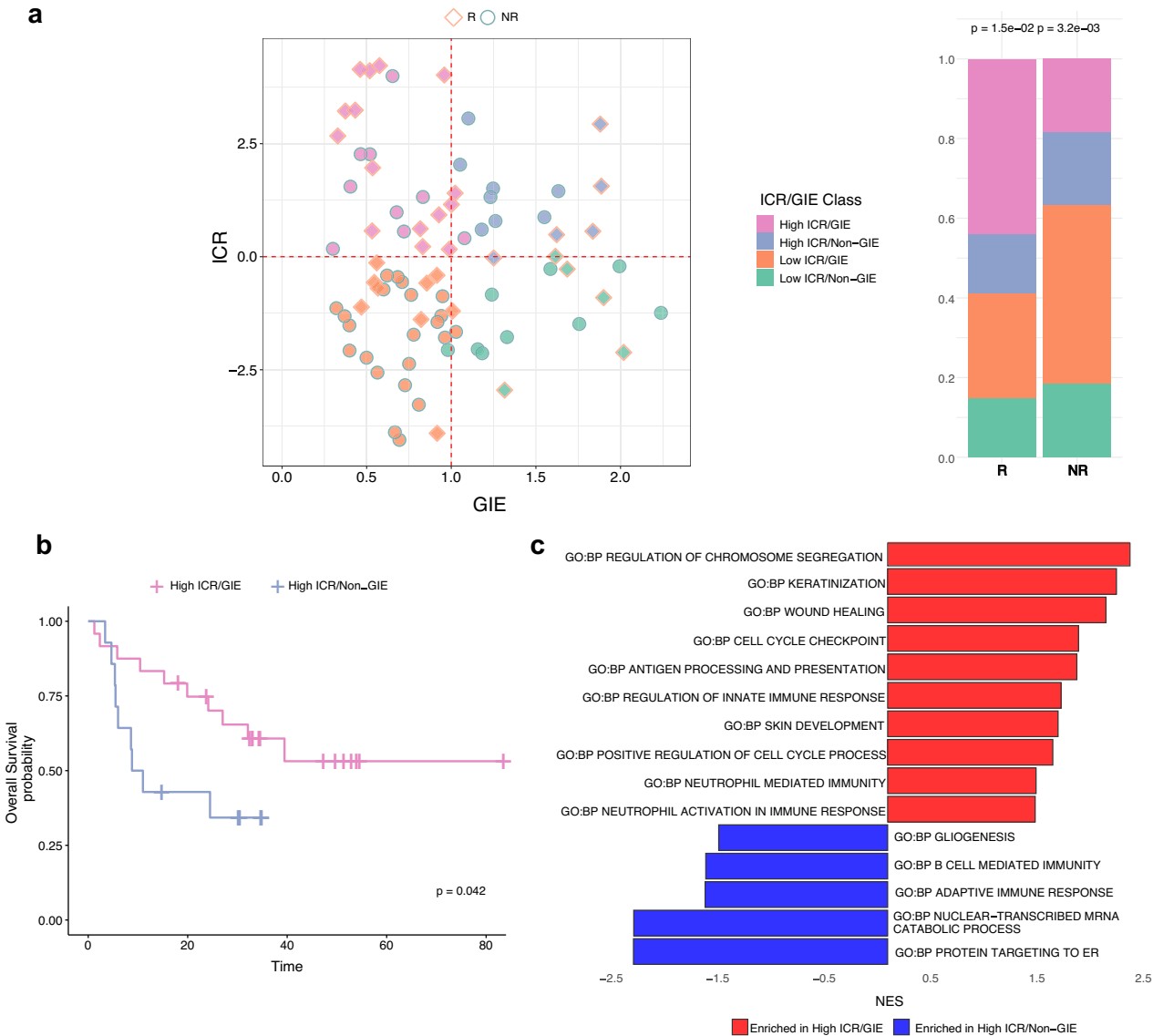

**Fig. 8 | Validation of the ICR/GIE score in patients from other cohorts.**
**a** Scatterplot of ICR score by Genetic Immunoediting (GIE) score for R ($n = 34$) and NR ($n = 49$) samples (left) and their proportion (right) after classification as ICR/GIE classes ($p$ value from two-sided Pearson's chi-squared test statistic).
**b** Kaplan–Meier for OS by patients classified as High ICR/GIE ($n = 24$) or High-ICR/ Non-GIE ($n = 14$). Time is indicated in months and censor points are indicated by vertical lines. $p$ value is calculated by log-rank test. **c** Barplot of most significantly (FDR < 0.01) enriched GO:BP terms from GSEA analysis of High-ICR/GIE ($n = 24$) vs. HighICR/Non-GIE ($n = 14$) comparison. Source data are provided as a Source Data file.

(NCT04250246) in PD-1/-L1-resistant melanoma and non-small cell lung cancer patients will help to address the clinical and immunobiologic contribution of the addition of the HMA decitabine/cedazuridine (ASTX727) to ICI therapy. However, further support for the notion that guadecitabine is a promising way forward to improve the efficacy of ICI therapy in solid tumors derives from two most recently published trials in platinum-resistant ovarian cancer[14], and in different tumor types[15]. Indeed, the combination of guadecitabine with the anti-PD-1 pembrolizumab led to encouraging response rates, immunomodulation in the tumor tissue and/or in periphery, and evidence of demethylation in on-treatment lesions, with manageable toxicity[18].

Our aim here was to exploit the longitudinal multi-omics profiling of the NIBIT-M4 cohort in conjunction with the 5 years follow-up to shed light on the effect of the combination during treatment and to identify early biomarkers of response. To this end, we have first individually interrogated the available omics platforms taking advantage of the accurate longitudinal sampling. Then we developed computational multi-omics integration approaches to evaluate the effect of the

adopted demethylation agent on boosting the adaptive and innate immune-mediated cancer rejection. The five-year follow-up showed that our multi-omics classification based on the ICR/GIE index is a better prognostic factor than best overall response (BOR). Therefore, the analysis reported here can serve as a guide for improved patient stratification and selection strategies in combination therapies involving ICI and immunomodulatory agents, including HMA.

Longitudinal WES and transcriptomic analysis contributed to shed light on molecular factors impacting clinical response and on long-term outcome after guadecitabine plus ipilimumab. At the mutational profile level, strong consistency across biopsies obtained at three-time points in a 12-week time frame allowed us to identify significant associations of somatic mutations with response/resistance even with the small number of patients enrolled in the NIBIT-M4 trial.

Even if the impact of the genomic landscape is difficult to be linked to the HMA rather than to ipilimumab, our results confirm and extend evidence of resistance obtained in ICB-only trials. First, we

found that *BRAF* mutations are more frequent in NR compared to R patients and the opposite was true for *NRAS* mutations. Patients with *NRAS* mutation-positive melanomas have improved PFS after ICB with anti-CTLA-4 and -PD-1 antibodies[22] and improved OS when treated with ipilimumab[23]. Concerning *BRAF* mutations, there are conflicting reports in the literature[27]. Some meta-analyses did not evidence differences in response studies between *BRAF* wild-type and mutation patients[53]. Other studies reported a negative association between *BRAF* status and OS after ipilimumab in melanoma[54]. Thus, the associations that we found with *BRAF* and *NRAS* mutational status, with the caveat of the limited sample size in our trial, appear consistent with evidence obtained in clinical trials of ICB-only. These data thus suggest that the associations of *BRAF/NRAS* mutations with clinical response may reflect their impact on the ICI, rather than their relevance for the mechanism of action of the demethylating agent. We found the *CDKN2A* mutations to be more frequent in NR patients. *CDKN2A* alterations have been previously found associated with reduced benefit from ICB in urothelial carcinoma, but not in melanoma[24]. Immunotherapy resistance of *CDNKA* deficient tumors has also been reported in pan-cancer studies[55]. *EPHA7* mutations, which we found more frequently in R patients, have been associated with better clinical outcomes in ICB-treated patients across multiple cancer types[25]. *PLXNA4* mutations, more frequent in R patients, have been shown to promote Cytotoxic T-lymphocyte infiltration in pre-clinical tumor models, as *PLXNA4* behaves as a negative checkpoint regulating T-cell migration and proliferation[26].

Mutations in genes belonging to the EMT were enriched in NR patients. EMT is associated with a less favorable outcome in patients with cancer[56]; it stimulates angiogenesis and is a tumor-intrinsic mechanism enhancing immunosuppression[57,58]. Recent studies reporting the genomic and transcriptomic features associated with ICI in melanoma have shown higher expression of several EMT genes in NR subjects[59]. Our analysis indicates that genomic alterations can drive these differences. We also observed that two NR patients harbored mutations in the gene *DNMT1*, which is the direct target of guadecitabine. We described the reduced immune-modulatory effect of the demethylating agent in patients harboring defects. In summary, available evidence indicates that most differentially enriched mutations between R and NR groups have also been associated with response or resistance to ICB-only regimens. In other words, the genomic landscape of the lesions may contribute to explaining the efficacy or lack of efficacy of the ipilimumab arm of the trial.

Interrogating the longitudinal gene expression data was also useful in evaluating how the immune context evolves during therapy. The differential activity of these pathways tends to increase during therapy, suggesting that immune surveillance promoted by ICI represses cell cycle genes together with differentiation pathways. We have shown an increased expression of the guadecitabine-specific signature that we had previously defined[10] in on-treatment lesions from R compared to NR patients. This result suggests that clinical benefit requires susceptibility to the immunomodulatory action of this DNMT inhibitor. A consequence of this hypothesis is that NR patients must have some resistance mechanism to guadecitabine, and the *DNMT1* mutations found in two NR patients may be part of this resistance mechanism.

The transcriptional effects that could be ascribed to guadecitabine and could contribute to explain clinical benefit in this combinatorial trial, were assessed by considering tumor-specific antigens derived from transposable elements[60] and ERV sequences[28] for the immune response to tumors. Both these classes of sequences are known to be regulated by methylation and susceptible to re-activation by demethylating agents. We confirmed that clinical benefit in the NIBIT-M4 trial could depend on the ability of guadecitabine to re-activate these sequences, by promoting their demethylation, leading to enhanced expression in R compared to NR patients.

We evaluated several transcription-based signatures for response prediction to ICI. No difference was observed at the baseline. This suggests that additional factors contribute to clinical response, beyond the process of development of adaptive immunity, captured by these signatures. We reasoned that an immune-based stratification approach taking into account: a) the evidence of expression of genes associated with ICR and b) the amount of immunoediting measured as the ratio of observed versus expected neoantigens (GIE) could improve our ability to understand response and resistance to treatment. The presence of an adaptive immune response within tumors is accounted by the ICR[16,18], a signature that predicts survival and response to ICI therapy in different tumors such as breast[61], bladder, stomach, head and neck[16], sarcoma[20], and melanoma[8]. The importance of immunoediting, and its association with survival and resistance has been extensively demonstrated in human primary tumors[17,49] and in immune selection pressure on metastatic evolution[62]. The GIE score is a measure of the extent of genetic immunoediting occurring in a tumor and it is obtained through comparison of expected number of neoantigens (through a linear model relating mutational load and neoantigen load) with the observed number of neoantigens (i.e., actual number of neonatigens observed in a specific patient). Patients whose tumors have a GIE score <1 indicate previous immunoediting. The observation that the GIE score and the ICR were uncorrelated suggested that they are capturing complementary, yet distinct, attributes of anti-tumoral immunity. For instance, an antitumor immune response can occur against non-mutated antigens[63]. Therefore, their combination could be an effective means to achieve a more accurate quantification of effective cancer immune surveillance and of response to immunotherapy. Indeed, the ICR/GIE classification could stratify NIBIT-M4 patients into four subsets, and those with high (>0) ICR scores and a low (<1) GIE score (truly immunoedited lesions) showed the longest OS. In other words, the coupling of adaptive immunity with effective immunoediting is a relevant requisite for achieving long-term clinical benefit from treatment. The prognostic significance of the ICR/GIE index was confirmed by the analysis of independent datasets from ICI-treated patients, suggesting that this is a robust classifier that captures crucial immunological processes acting in the context of different immunotherapy regimens. Moreover, its prognostic value has been recently validated also in a large cohort of colon cancer[17]. The activation of T-cell-mediated immunity is dependent upon the recognition of tumor antigens on MHC of antigen-presenting cells[64]. Tumor antigen presentation by MHC class I is mediated by the coordinated expression of multiple genes. The differences between the High-ICR/GIE and High-ICR/Non-GIE, observed in our cohort and other independent cohorts, confirm that even in the presence of an adaptive immune response, tumor cells that develop defects in antigen processing or presentation can escape immune surveillance[48,65]. In fact, lesions with defective expression of HLA class I molecules may retain evidence for the development of adaptive immunity (High-ICR) and also for CD8+ T-cell infiltration, but down-modulation of MHC class I molecules on tumor cells prevents recognition of HLA/neoantigen complexes by T cells, thus suppressing the possibility of genetic immunoediting (therefore, the lesions are identified as Non-GIE). We have previously shown that DNA methyltransferase inhibitors can upregulate the expression of MHC proteins[43]. We hypothesize that the HMA treatment might improve the response to ICI, specifically in melanoma patients with high ICR scores but lower evidence of immune editing. Moreover, serial monitoring of these relative scores might show increased expression of MHC class I in the DNMTi-treated samples.

The inference of changes in gene expression programs needs to account for the cellular composition at different time points. The actual profiling platform, with half of the cases lacking the normal reference, is not ideal for an accurate estimation of the purity. Single-cell profiling will be performed on this cohort in future studies to

address this point. Moreover, the low number of analyzed cases, even with longitudinal multi-omics profiling, represents a limitation in our study, though its results help to guide further development of HMA/ICI combinations in solid tumors. Additionally, the application of the ICR/GIE to multiple contexts at a pan-cancer level is needed to further validate the clinical relevance of our approach and findings. Collectively, though limited by the still scarce number of completed trials and enrolled patients, the available clinical results suggest and support further development of combinatorial approaches of HMA with ICI in cancer therapy in randomized clinical trials with extensive translational endpoints.

## Methods

### Study design, patient population, procedures, and outcomes
The phase Ib NIBIT-M4 study (NCT02608437) was conducted in accordance with the ethical principles of the Declaration of Helsinki and the International Conference on Harmonization of Good Clinical Practice. The protocol was approved by the independent ethics committee of the University Hospital of Siena (Siena, Italy). All participating patients (or their legal representatives) provided signed informed consent before enrollment.

We conducted a milestone, 5-year follow-up analysis of patients enrolled in the NIBIT-M4 study; the study design, patient eligibility criteria, and treatment regimen have already been described[11]. Briefly, the phase Ib, dose-escalation, single-center NIBIT-M4 study, enrolled pre-treated or untreated patients with unresectable Stage III or IV melanoma, to receive guadecitabine 30, 45, or 60 mg/m²/day s.c. on days 1–5 at week 0, 3, 6, 9, and ipilimumab 3 mg/kg i.v. on day 1 at week 1, 4, 7, 10, for 4 cycles. For this follow-up analysis, median OS, PFS, 5-year OS PFS rate, and median DoR were assessed. Patients were classified as R if they experienced a DC [defined as CR, Partial Response (PR), or SD]. Patients who experienced a progressive disease (PD) were classified as NR. Tumor biopsies for correlative analyses were performed at baseline and at week 4 and week 12 on-treatment.

### Data collection, library preparation, and sequencing
Isolation of total DNA/RNA and library preparation for RNA Sequencing and RRBS were performed as previously described[11] at different time points of treatments (week 0, week 4, week 12) for $N = 14$ patients, including eight additional patients not available in the previous study. For WES, Nextera Flex for Enrichment solution (Illumina, San Diego, CA) in combination with SureSelect Human All Exon V7 probes (Agilent, Santa Clara, CA) was used for library preparation and generated libraries were sequenced on NovaSeq 6000 (Illumina, San Diego, CA) in 150 pair-end mode for biopsies of patients from 1 to 8; TruSeq Rapid Exome (Illumina, San Diego, CA, USA) was used for library preparation and generated libraries were sequenced on HiSeq 3000/4000 (Illumina, San Diego, CA) in 150 pair-end mode for biopsies of patients from 9 to 14.

### Data processing
**Whole exome sequencing.** Quality control of WES was performed on raw data using fastQC (v. 0.11.8) (https://www.bioinformatics.babraham.ac.uk/projects/fastqc/).

Sequencing reads were aligned to the Human reference genome (UCSC genome assembly GRCh37/hg19) using Burrows-Wheeler Aligner[66], and then processed by GATK[67] for discarding low mapping quality reads and performing indel realignment.

Somatic single-nucleotide variants (SNVs) and indels calling were performed using Sentieon Genomic Tool v. 201911[68]. A virtual normal panel from 1000 Genomes Project[69] was used to call SNVs and indels for tumor samples without a matched normal sample.

Putative false positive calls have been removed considering as filters: (i) the variant-supporting read count greater than 2; (ii) variant allele frequency greater than 0.05; (iii) average variant position in variant-supporting reads (relative to read length) >0.1 and lower than 0.9; (iv) average distance to effective 3′ end of variant position in variant-supporting reads (relative to read length) greater than 0.2; (v) fraction of variant-supporting reads from each strand >0.01; (vi) average mismatch quality difference (variant−reference) lower than 50; vii) average mapping quality difference (reference−variant) lower than 50. Annotation of SNVs and indels was performed using AnnoVar[70] and SnpEff[71]. The functional effect of missense SNVs and in-frame indels was computed using Polyphen2[72], SIFT[73], and PROVEAN[74] algorithms and variants predicted as damaging at least two of them were classified as pathogenic mutations. Somatic copy number was estimated from WES reads by CNVkit[75] and GISTIC[76] was applied for identifying genomic regions recurrently amplified or deleted. The nonsynonymous tumor mutational burden (TMB) was computed as the number of nonsynonymous somatic mutations (single-nucleotide variants and small insertions/deletions) per megabase in coding regions. COSMIC mutational signature v3.2[77] frequencies were computed using deconstructSigsR package (v. 1.8.0)[78] for each tumor sample derived from a patient where a matched normal sample was available ($n = 8$).

**RNA sequencing.** Fastq quality was assessed using fastQC (v. 0.11.8) (https://www.bioinformatics.babraham.ac.uk/projects/fastqc/) and low-quality reads were discarded. Sequence reads were aligned to Human reference genome (UCSC genome assembly GRCh38/hg38) using STAR (v. 2.7.0b)[79], and the expression was quantified at gene level using featureCounts (v. 1.6.3), a count-based estimation algorithm[80]. Downstream analysis was performed in the R statistical environment as described below. Raw data were normalized according to sample-specific GC-content differences as described in EDAseq R package (v. 2.22.0)[81]. Differential expression analysis was performed using EdgeR R package (v. 3.30.3)[82]. Genes sorted according to log2 fold-change (log2FC) were used for performing Gene Set Enrichment Analysis (GSEA) of Gene Ontology (GO) Biological Processes (BP)[83], as implemented in the clusterProfiler R package (v. 3.3.6)[84]. For the expression quantification of genomic repetitive DNA features, *featureCounts* function of the Rsubread R package (v. 2.10.5) was applied by enabling the parameters of useMetaFeatures and countMultiMappingReads and using a GTF file built on repetitive DNA feature localizations as described below. Any genomic repetitive DNA features that overlapped with GRCh38 exonic regions were excluded from the analysis. For each feature type, the resulted multi-mapping reads weighted by the number of mapping sites were first averaged and then log2 transformed.

**RRB sequencing.** RRBS raw reads were trimmed for adaptor sequences using trim galore (v. 0.6.5) (http://www.bioinformatics.babraham.ac.uk/projects/trim_galore/) and filtered for low-quality sequences using fastQC (v. 0.11.8) (https://www.bioinformatics.babraham.ac.uk/projects/fastqc/). High quality trimmed reads were mapped to the Human reference genome (UCSC genome assembly GRCh38/hg38) using Bismark (v. 0.22.3)[85] with default parameters. Methylation data as β values for CpG sites, promoters, and genes were retrieved from bismark coverage outputs using R package RnBeads 2.0 (v. 2.6.0)[86] with default parameters. Then, human GRCh38 annotated genes and promoters exhibiting differential DNA methylation between pre-defined groups of patient samples were identified using R package limma (v. 3.44.3)[87].

The methylation of the considered genomic feature was computed by first overlapping the CpG positions with GRCh38 genomic feature localizations using GenomicRanges R package (v. 1.48.0)[88] and then averaging methylation β values of single CpGs in each feature. Only CpG sites outside regions annotated as "Open Sea" and with a minimum average of 10× coverage depth across all samples were selected for the analysis. Genomic feature localizations (exon,

intergenic regions, intron, promoter, and UTRs) were retrieved from UCSC (genome assembly GRCh38/hg38) using AnnotationHub R package (v. 3.4.0). Regulatory feature localizations were retrieved from the following public resources: enhancer features from Gene-Hancer Version 4.4[89] within the GeneCards® Suite (https://www.genecards.org/), super-enhancer features from H3K27Ac ChIP-seq data of GEO Dataset GSE99835[90] and CTCF binding site features from CTCF ChIP-seq data of Geo Dataset GSE128346[91]. The original GRCh37/hg19 enhancer and super-enhancer feature localizations were lifted over GRCh38/hg38 genomic positions using liftOver R package (v. 1.20.0).

Repetitive DNA feature localizations (Long interspersed nuclear elements, LINE; Short interspersed nuclear elements, SINE; and Long terminal repeat elements, LTR) were retrieved from UCSC Repeat-Masker annotation (genome assembly GRCh38/hg38) using AnnotationHub R package.

To investigate associations between CpG site-specific DNA methylation of genomic features and clinical response, robust linear regression models were fitted at each treatment time point for considered patients' groups using the MASS R package (v. 7.3–58.3).

Human GRCh38 annotated genes and promoters exhibiting differential DNA methylation between predefined groups of patient samples were identified using R package limma (v. 3.44.3)[87].

## Prediction of immune response and tumor microenvironment deconvolution

ICR scores were computed for each sample using a single-sample GSEA (ssGSEA) based on the Mann–Whitney–Wilcoxon Gene Set Test (MWW-GST)[92] and the ICR signature (*IFNG, IRF1, STAT1, IL12B, TBX21, CD8A, CD8B, CXCL9, CXCL10, CCL5, GZMB, GNLY, PRF1, GZMH, GZMA, CD274/PD-L1, PDCD1, CTLA4, FOXP3,* and *IDO1*)[19].

MIRACLE scores were computed using the MIRACLE R package as described in Turan et al.[8]. TIDE score was computed using TIDE command-line interface (https://github.com/jingxinfu/TIDEpy)[93,94]. IMPRES score was computed using *calc_impres* R function (https://github.com/Benjamin-Vincent-Lab/binfotron/)[95].

Melanocytic plasticity signature (MPS) score was computed as described in Pérez-Guijarro et al.[35].

Estimation of immune and stromal subpopulation abundances was computed using MCP-counter[45].

## TCR repertoire analysis from RNA-sequencing data

The docker implementation of MiXCR software (v. 3.0.13)[96] was used to retrieve the VDJ repertoire from RNA-sequencing data. For the T cell receptor Beta locus (TRB), the clonality was calculated as

$$Clonality_{TRB} = 1 - \frac{1}{log_2 N} H(x) \qquad (1)$$

where $H(x)$ is the Entropy computed as standard Shannon entropy as follow:

$$H(x) = -\sum_{i=1}^{N} P(x_i) log_2 P(x_i) \qquad (2)$$

For a productive (in-frame) sequence $x_i, P(x_i)$, is the ratio between the sequence count and total productive count and $N$ is the number of productive unique in-frame sequences.

## Integrative analysis of RNASeq and RRBS data

Integration of gene expression and methylation data was performed as follows. R package SMITE (v. 1.16.0)[46] was applied to identify functionally related genes with altered DNA methylation on promoters. Briefly, each differentially expressed gene (R vs. NR patients at each treatment time point week 0, week 4, and week 12) previously computed using the EdgeR R package (v. 3.30.3)[82] was associated with a

promoter region [TSS − 1 kb, TSS + 500 bp] using UCSC GRCh38/hg38 refSeq transcripts coordinates. Each promoter was then associated with a set of overlapping regions from the differential methylation analysis previously computed on the same comparison using R package limma (v. 3.44.3)[87]. To identify genes whose expression is inversely correlated with promoter methylation, a score based on a weighted significance value (0.5 for expression and 0.3 for promoter methylation) was computed. Hypomethylated/upregulated and hypermethylated/down-regulated genes for each comparison were then visualized as scatterplot using R package ggplot2 (v. 3.3.6). These modules of expression/methylation concordant genes were functionally analyzed through a GO BP pathway enrichment analysis within the R package SMITE. Significant categories (*p* value < 0.05) were visualized as barplot using R package ggplot2.

## HLA typing and neoantigen prediction

HLA typing was performed from WES data using the docker implementation of Polysolver (v. 4)[97]. The neoantigen prediction tool pVACseq from pVACtools[98] was run using the following predictors: MHCnuggetsI, NNalign, NetMHC, SMM, SMMPMBEC, and SMMalign.

Mutant-specific binders, relevant to the restricted HLA-I allele, are referred to as neoantigens, as previously described[99]. To infer neoantigens with high confidence, we considered only the mutated epitopes with a median IC$_{50}$ binding affinity across all prediction algorithms used <500 nM, with a corresponding wild-type epitope with a median IC50 binding affinity >500 nM, and with at least one supporting read on the RNASeq data.

## Genetic ImmunoEditing (GIE) score

The Genetic ImmunoEditing (GIE) score was computed by taking the ratio between the number of observed neoantigens (*O*) in a patient versus the number of expected neoantigens (*E*) for that patient:

$$GIE = \frac{O}{E} \qquad (3)$$

The observed number of neoantigens *O* was obtained from the output of pVACtools[98], filtered according to the criteria described above.

The expected number of neoantigens *E* was computed as a function of the number of nonsynonymous mutations by fitting a linear regression model using the *lm* function of R package stats (v. 4.0.2) trained with the data of our cohort, using the number of neoantigens as dependent variable. Due to the presence of hypermutated samples, first we classified patients' samples into two groups, hyper- or hypomutated, according to a TMB cutoff of 12 mutations/mb, and then we fitted for each group a linear regression model.

We assumed that samples that show a frequency of neoantigens lower than expected (i.e., lower GIE values) have evidence of immunoediting.

Since the GIE is a ratio between observed and expected neoantigens, we used "GIE" or "Non-GIE" definitions to nominate samples with GIE < 1 or GIE > 1, respectively. Analogously, we adopted the normalized enrichment score (NES) to evaluate the activation of the ICR signature at the single-sample level using the yaGST tool[92]. A value of NES > 0 means positive activation, whereas a value of NES < 0 means a significant negative activation. Accordingly, these two conditions were used to nominate "High-ICR" versus "Low-ICR", respectively.

## Survival analysis

Survival curves were estimated using the survival R package (v. 3.2–10) and plotted using the Kaplan–Meier method, implemented in the survminer (v. 0.4.9) R package. Log-rank tests were used to compare curves between groups.

## IHC analysis

Serial 3 μm formalin-fixed paraffin-embedded tissue sections were stained using an AutostainerPlus (Dako). Antigen retrieval and deparaffinization were carried out on a PT-Link (Dako) using the EnVision FLEX Target Retrieval Solutions (Dako). Endogenous peroxidase and non-specific staining were blocked with H202 3% (Gifrer, 10603051) and Protein Block (Dako, X0909) respectively. The antibodies used are: anti-CD8 clone C8/144B (M7103, Dako) at a final concentration of 3.14 μg/mL and anti-HLA Class I, clone EMR8-5 (ab70328, abcam) at a final concentration of 0.3 μg/mL. The HRP-labeled polymer conjugated EnVision+ Single Reagent (Dako, K4001) was used as a secondary antibody. Peroxidase activity was detected using 3-amino-9-ethylcarbazole substrate (Vector Laboratories, SK-4200). All stained slides were digitalized with a Nano-Zoomer scanner (Hamamatsu).

## NanoString

Expression of genes belonging to several immune-related signatures was assessed by a custom-designed NanoString nCounter multiplex CodeSet enabling the determination of 364 genes. The gene signatures were selected for providing information on B-cell content and differentiation, TLS formation, follicular T helper cells, TEX subsets, tumor-associated endothelial cells, ICB response, and guadecitabine-specific gene upregulation. For NanoString experiments, panel probes (capture and report) and 200 ng of RNA were hybridized overnight at 65 °C for 16 h. Samples were scanned at maximum scan resolution capabilities (555 FOV) using the nCounter Digital Analyzer. Quality control of samples, data normalization, and data analysis were performed using nSolver software 4.0 (Nano-String Technologies).

## ICR/GIE validation in external cohorts

Molecular and clinical data for three independent immunogenomic datasets of melanoma patients treated with ICI[5,50,51] were obtained from cBioportal[100]. A total of 83 patients for which all required data were available (gene expression, mutation/neoantigen loads, treatment response, and overall survival) were selected and grouped into responder (CR, PR, SD, LB: long-term benefits) and non-responder (PD, NB: minimal or no-benefits) according to the treatment outcome as described in the corresponding original studies. The integrated gene expression matrix was batch-corrected using the *removeBatchEffect* function implemented in R package limma (v. 3.44.3)[87]. ICR and GIE scores were computed as previously described. Differential expression analysis was performed between "High-ICR/GIE" and "High-ICR/Non-GIE" classes using a Wilcoxon test. Genes sorted according to log2 fold-change (log2FC) were used for performing Gene Set Enrichment Analysis (GSEA) of Gene Ontology (GO) Biological Processes (BP)[83], as implemented in the clusterProfiler R package (v. 3.3.6)[84]. Selected enriched GO:BP terms (FDR < 0.01) were visualized as a barplot. Survival analysis was performed as previously described.

## Power analysis

To confirm the statistical power to detect a significant difference between R and NR patients, we conducted a post hoc power analysis using sample size and a hypothesis of effect size to detect. We used in all our comparisons a significance level of 0.05 and assumed a two-tailed test. The reported comparison includes the differential analyses at each time point between R and NR patients using both gene expression and methylation profiling. With an estimated guess of effect size (Cohen's *d*) of 1.5 and a sample size of 14 (8 in NR and 6 in R patients), the post hoc power analysis revealed that our study had a statistical power of 72% to detect a significant difference between the two groups. The low number of cases of course is a major limit of the detectable effect size.

## Reporting summary

Further information on research design is available in the Nature Portfolio Reporting Summary linked to this article.

## Data availability

The processed NanoString data generated in this study has been deposited in the GEO under accession number: GSE211645. Raw data of WES, RNA-sequencing, and RRBS are available under restricted access, for privacy of the patients, on the European Genome-phenome Archive under accession number EGAS00001006736. Access for research purposes can be obtained by applying to the data access committee via EGA (see: EGAC00001002947). It is expected that data will be available within 3 months of the request and there are no restrictions on the duration of access. The complete de-identified clinical data are available under restricted access. Data access can be obtained by request from segreteria@fondazionenibit.org. The data that will be shared include individual participant data that underlie the results reported in this paper after de-identification (text, table, figures, and appendices). The time frame for response to requests will be within four weeks. Data will be shared for non-commercial purposes after approval of a proposal by the Board of the NIBIT Foundation and with a signed data access agreement. The availability of such data will begin 3 months and end 24 months after article publication. The study protocol is available as a Supplementary Note in the Supplementary Information file. Previously published genomic data of independent cohorts used to support the findings of this study were obtained from the CBioPortal for Cancer Genomic (https://www.cbioportal.org/study/summary?id=mel_ucla_2016, https://www.cbioportal.org/study/summary?id=mel_dfci_2019, https://www.cbioportal.org/study/summary?id=skcm_mskcc_2014) and from GEO under Series accession number: GSE188250. The remaining data are available within the Article, Supplementary Information, or Source Data file. Source data are provided with this paper.

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

## Acknowledgements

The research leading to these results has received funding from Fondazione AIRC under 5 per Mille 2018—ID.21073 project—P.I. Maio Michele, G.L. Anichini Andrea, G.L. Ceccarelli Michele and from AIRC under IG 2018—ID. 21846 project—P.I. Ceccarelli Michele. This work was also supported by the Italian Ministry of Research Grant PRIN 2017XJ38A4_004 and from the Ministry of Health, Lombardy and Tuscany regions, Bando Ricerca Finalizzata, grant number NET-2016-02361632, Program P.I. Michele Maio, P.I. work package 2 Andrea Anichini. We thank Astex Pharmaceuticals Inc. that provided the study drug guadecitabine.

## Author contributions

Conceptualization: T.M.R.N., A.M.D.G., A.A., M.M. and M.C.; methodology: T.M.R.N., A.M.D.G., W.H.F., C. S.-F., R.M., M.F. L., D.B., A.A., M.M., and M.C.; investigation: T.M.R.N., A.M.D.G., F.P.C., A.C., S.C., W.H.F., C. S.-F., S.B., G.P., E.S., M.M., and M.C.; analysis: T.M.R.N., A.M.D.G., F.P.C., A.C., G.S., M.C.C., R.T., S.C., W.H.F., C. S.-F., S.B., G.P., M.M. and M.C.; writing: T.M.R.N., A.M.D.G., R.M., A.A., M.M. and M.C; supervision: M.M. and M.C.

## Competing interests

A.M.D.G. has served as consultant and/or advisor to Incyte, Pierre Fabre, Glaxo Smith Kline, Bristol-Myers Squibb, Merck Sharp Dohme, and Sanofi and has received compensated educational activities from Bristol-Myers Squibb, Merck Sharp Dohme, Pierre Fabre, and Sanofi. W.H.F. has served as consultant and/or advisor to AstraZeneca, Adaptimmune, Catalym, OOSE Immunotherapeutics, and Novartis, and reports receiving speakers bureau honoraria from Bristol-Myers Squibb. MM has served as consultant and/or advisor to Roche, Bristol-Myers Squibb, Merck Sharp Dohme, Incyte, AstraZeneca, Amgen, Pierre Fabre, Eli Lilly, Glaxo Smith Kline, Sciclone, Sanofi, Alfasigma, and Merck Serono; and own shares in Theravance and Epigen Therapeutics, Srl. MC serves as consultant and/or advisor to Moderna Therapeutics and is the founder and owns shares of Immunomica srl. Other authors have nothing to declare.

## Additional information

Teresa Maria Rosaria Noviello [1,2,17], Anna Maria Di Giacomo[3,4,5,17], Francesca Pia Caruso [2,6], Alessia Covre[3], Roberta Mortarini[7], Giovanni Scala[8], Maria Claudia Costa [2,6], Sandra Coral[3], Wolf H. Fridman [9,10,11], Catherine Sautès-Fridman [9,10,11], Silvia Brich[12], Giancarlo Pruneri[12], Elena Simonetti[4], Maria Fortunata Lofiego[3], Rossella Tufano[2,13], Davide Bedognetti [14,15], Andrea Anichini [7], Michele Maio [3,4,5,18] ✉ & Michele Ceccarelli[1,16,18]

[1]Sylvester Comprehensive Cancer Center, Miller School of Medicine, University of Miami, Miami, FL, USA. [2]BIOGEM Institute of Molecular Biology and Genetics, Ariano Irpino, Italy. [3]University of Siena, Siena, Italy. [4]Center for Immuno-Oncology, University Hospital of Siena, Siena, Italy. [5]NIBIT Foundation Onlus, Siena, Italy. [6]Department of Electrical Engineering and Information Technology (DIETI), University of Naples "Federico II", Naples, Italy. [7]Human Tumors Immunobiology Unit, Dept. of Research, Fondazione IRCCS Istituto Nazionale dei Tumori, Milan, Italy. [8]Department of Biology, University of Naples "Federico II", Naples, Italy. [9]INSERM, UMR_S 1138, Centre de Recherche des Cordeliers, Team Cancer, Immune Control and Escape, Paris, France. [10]University Paris Descartes Paris 5, Sorbonne Paris Cite, UMR_S 1138, Centre de Recherche des Cordeliers, Paris, France. [11]Sorbonne University, UMR_S 1138, Centre de Recherche des Cordeliers, Paris, France. [12]Department of Pathology and Laboratory Medicine, Fondazione IRCCS Istituto Nazionale dei Tumori, Milan, Italy. [13]Department of Science and Technology, University of Sannio, Benevento, Italy. [14]Cancer Program, Human Immunology Department, Research Branch, Sidra Medicine, Doha, Qatar. [15]Department of Internal Medicine, University of Genoa, Genoa, Italy. [16]Department of Public Health Sciences, Miller School of Medicine, University of Miami, Miami, FL, USA. [17]These authors contributed equally: Teresa Maria Rosaria Noviello, Anna Maria Di Giacomo. [18]These authors jointly supervised this work: Michele Maio, Michele Ceccarelli. ✉e-mail: maio@unisi.it

