## [Peer Review File · Nature Communications]

Guadecitabine plus ipilimumab in unresectable melanoma:
five-year follow-up and integrated multi-omic analysis in the
phase 1b NIBIT-M4 trialREVIEWER COMMENTS

Reviewer #1 (Remarks to the Author): with expertise in in bioinformatics, cancer immunology

Major comments

(1) “Genomics landscape: mutational profile differences in baseline and on-treatment lesions in R vs. NR patients” In this section, the authors observed a lot of differences. But the authors fail to link these differences in R vs NR to the mechanism of response/resistance, for DHA alone or ICI alone, or for their synergistic effects. Besides, the somatic mutations being consistent across tumor biopsies of the same patients across different time points suggest a lack of tumor evolution during treatment. This means the DHA and/or ICI treatment is not very effective? If not, what does it mean? The authors need to interpret what they observed.

(2) “Transcriptional landscape of baseline and on-treatment tumor lesions: distinct and evolving transcriptional programs distinguish R from NR patients” In this section the authors talked a lot about the differences in gene expression in the immune angle, which makes sense. But the authors didn’t mention anything about DHA. Does the difference in gene expression between R and NR reflect differential response to DHA (and maybe in combination with ICI)?

(3) “By a custom-designed NanoString assay we then explored differential expression in R vs NR lesions of ...” I don’t understand why the authors need to resort to NanoString here. Can’t the RNA-seq data also provide this same information? Or should the authors do the same analyses for both nanostring and RNA-seq and check if concordant results show up?

(4) “Integrative analysis of methylation and transcriptomic profiles during treatment” The authors did little analyses with respect to methylation and spent very little time to explain their analyses. But this is extremely important. With the current form of the manuscript, I cannot say I believe in the author’s analyses in this part.

a) “A rank aggregation analysis of gene expression and gene methylation in NIBIT-M4 tumor biopsies demonstrated the interdependent changes between gene body methylation levels and gene expression induced by guadecitabine” What part of what figure are you talking about? Need a lot more details about how you analyzed your data. Also, according to this description,

your figure and figure legend, you intersected the differences observed in methylation data and RNA expression data, and just ran some GO analyses. This doesn't show "interdependent changes" at all. This statement is wrong. It is not like you are investigating the correlation between methylation and RNA expression

b) "By exploiting further the expected inverse relationship between gene expression and promoter methylation, we then identified modules of hypomethylated, immune-related genes up-regulated during treatment in R vs NR patients (shown in the upper right quadrant in each plot of Figure 6B)." I am lost here. How did you "exploit"? How did you identify the modules? The exact analytical procedure?

(5) "By stratifying patients according to the ICR/GIE classification of week 12 biopsies we found a significant difference (p-value 0.035) in overall survival (OS) and in PFS between the High ICR/GIE and the High ICR/Non-GIE group (Figure 7C and Supplementary Figure 4B, left hand panel)." The number of patients here is very small. I am not sure how reliable the p value is, even if it is significant. Also the presentation of Fig 7c is questionable. The censored patients need to be shown with "+" on the curves. None of the patients were censored?

(6) "To validate the ICR/GIE stratification, we assembled a cohort of 83 melanoma cases treated with ICI, either anti-CTLA4 or -PD1, from previous published studies^{4,39,40} for which TMB, neoantigen load and gene expression were available" I am not sure how helpful this analysis is. These patients were treated with ICI only, but the authors' own patients were treated by ICI and DHA. In fact, this paper's biggest selling point is the combination therapy and now the authors seem to be backpadding on their own narrative.

(7) "This analysis confirmed the stratification of patients into 4 subsets as seen in the NIBIT-M4 cohort (Figure 9A)." This statement is simply wrong. This figure didn't confirm the existence of 4 subsets in the patients, as there is no clear bimodal distribution on the GIE axis, or on the ICR axis. The 4 subsets the authors identified were forced unnaturally.

(8) Overall, the genomics analyses in this paper rely too much on gene ontology analyses. The authors need to perform different types of genomics analyses from different angles, to cross validate the same story. Ideally, some orthogonal experiments should be performed to validate

key elements of their story. At the very least, GO analyses cannot distinguish the direction of gene regulation. For example, a GO term could have genes that are supposed to be up- or down-regulated in a certain condition. The authors may input a list of genes that are all up-regulated, which show a big overlap with this GO term. But those genes that are supposed to be down-regulated are in fact up-regulated, making this GO term a false positive. Many of the authors' figures show only simple GO analyses, without any other orthogonal analyses to rule out these issues, making them questionable.

Minor comments

(1) "Previous studies have shown that genetic evidence of neoantigen depletion is preferentially observed in lesions with high Immunoscore". Need to explain what is "immunoscore" and add a citation for it. "while progressing clones tend to be immune privileged" Same for "immune privileged"

(2) "The presence of an adaptive immune response within tumors is accounted for by the Immunological Constant of Rejection (ICR) 19,20." Need one sentence in this paragraph, immediately following this sentence, to explain what ICR really measures and what it means

(3) A major drawback of this study is the small number of patients. But I understand there is not too much the authors can do about this and the current collection of data is already a significant effort. But I feel it might still be better to acknowledge this deficiency, especially in places where sample size is critical. This includes, for example, the section on tumor mutations (Fig. 2). The data are very sparse in the waterfall plot, and the "enrichment" of genes is really weak

Reviewer #2 (Remarks to the Author): with expertise in melanoma, clinical

This manuscript focuses on the clinical and translational results from an Italian cooperative group study utilizing ipilimumab with the demethylating agent guadecitabine in advanced melanoma patients. It is hypothesized that demethylating agents will work well with ICI as these agents are known have immunomodulatory effects on tumor cells and activate innate immune pathways. There is preclinical evidence supporting the combination therapy as well showing guadecitabine is able to promote gene expression of interferons and TLRs (known to be implicated in innate immune response).

The plan for this trial was to perform longitudinal multi-omics profiling including WES, RNAseq and reduced-representation bisulfite sequencing on tumor biopsies collected at baseline, week 4 and week 12 on 14 of the treated patients.

Key findings include:

The clinical outcome data was impressive for an ipi-based regimen with 5 year OS at 28.9% which is higher than the traditional 20% seen on single-agent ipi trials.

No difference in TMB in R vs NR. NRAS more common in R, BRAF more common in NR. EMT genes more common in NR. CDKN2A more in NR. Dynamic increase in level of CD8 and NK cells in responders was more important than the baseline immune cell composition.

Ongoing randomized phase 2 study to test efficacy of addition of ASTX737 to ICI in pd1 refractory melanoma.

The bulk of the manuscript was in describing the multi-omics data with very little provided in terms of clinical data as it has been previously reported. The work was original and the conclusions were supported by the provided data. There are no clear flaws in data analysis or interpretation and methodology is sound. The investigators should be applauded for collecting so many good quality biospecimens to allow such detailed analyses.

Were patients PD1 refractory or naïve?

Could you be more specific about categorizing patients as R or NR based on "Disease Control." How was disease control defined?

Are there any further studies planned with guadecitabine in melanoma or is the focus on the ASTX compound that is being used in the randomized phase 2 study?

Reviewer #3 (Remarks to the Author): with expertise in in cancer genomics, immunology

Summary: Noviello et al analyze matched pre- and on-treatment samples for the NIBIT-M4 clinical trial of unresectable melanoma treated with the combination of guadecitabine and ipilimumab. They analyze somatic variation and gene expression in responders and nonresponders across time points. Using a number of scores capturing characteristics of the tumor and tumor immune microenvironment, they find that no single score accurately predicts which patients will respond. However, a combination of two scores ICR and GIE which capture evidence of active immunity at the tumor site and immunoediting respective, is capable of predicting which patients are likely to respond. They validate this score in independent cohorts. They also note that responders tend to have increased immune signatures at later time points whereas non-responder tumors do not. Overall, there are few human studies with genomic and transcriptomic data from on-treatment samples, and the analysis presented here is likely to be of broad interest to the community. The ICR/GIE score seems promising as well. Overall the manuscript is clearly written. Certain aspects could use further clarification.

Major Comments:

It was not entirely clear how the authors compared across time points for specific analyses (gene expression, immune infiltration, methylation). For such analyses (e.g. Figure 3A, 5A), did the authors control for non-independence of different timepoints for the same? Can the authors quantify the effect size / significance of the increase between timepoints from in 3A? Right now, the reader is asked to accept the increase based on the visual representation, but the underlying calculation (i.e. enrichment for certain GO:BP gene sets over time) is complex so the interpretation is not entirely straightforward.

It was unclear to me whether the use of guadecitabine could be credited for aiding the immune response in responders. The authors note that in responders, there is increasing hypomethylation of immune genes over time relative to nonresponders. However, there are also higher expression levels of immune genes and higher levels of immune infiltrate. How do the authors rule out that a higher proportion of DNA coming from cells actively expressing those immune genes explains the change in methylation, rather than the drug.

Did the authors note changes to methylation from baseline in general? I.e. rather than perform differential methylation between R and NR at different time points, did the authors compare differential methylation within R and NR groups over time? What genes were affected in non-responders? Presumably non-response to an immunotherapy does not necessarily imply non-response to a demethylating agent, so one might expect to see some effects attributable to guadecitabine in the NR that may be more reflective of changes in methylation more generally. What genes were affected in the absence of a robust immune response?

Does GIE change if the authors limit to mutations that show evidence of RNA support (variant allele present in RNA reads)? One would expect that removing mutations that are not expressed would strengthen the signal if anything. Also, the motivation for the score cutoffs used to designate ICR_{high} and GIE_{high} was not entirely clear. Are the authors splitting on the median score? Since the basic cutoffs validate, could the authors try to see if learning a cutoff based on NIBIT-M4 improves separation in the validation cohort (maybe using a regression model?). Could the ICR/GIE score be combined with other covariates to improve prediction based on the pre-treatment sample?

Did the authors note a loss of DNA/RNA support for neoantigens over time in responder lesions relative to non-responders? This analysis may need to control for tumor content of the samples though, as immune cells will be more abundant in on-treatment samples, potentially dominating the DNA / RNA content of the sample.

The authors conclude from expression analysis that immune surveillance suppresses the cell cycle and differentiation pathways. This is one possibility. Did the authors evaluate whether the apparent reduction in these pathways could reflect that there are many more immune cells contributing RNA to the sample and thus it could be a relative effect? Could this be normalized to the tumor cell content of the samples?

One of the melanoma differentiation scores, MPS, was suggested to relate to mechanism of immune evasion. It might be interesting to look at MPS score as well: Pérez-Guijarro, E., Yang, H.H., Araya, R.E. et al. Multimodel preclinical platform predicts clinical response of

melanoma to immunotherapy. Nat Med 26, 781–791 (2020).

<https://doi.org/10.1038/s41591-020-0818-3>

It was not clear whether multiple testing correction was performed or what methods was used to do so.

Minor Comments:

Consider using lower case r to denote pearson's correlation coefficient to avoid confusing with R indicating the responders.

Why are the $-\log_{10}p$ values in the bottom half of 3A negative but positive in the top half? Shouldn't p-value be independent of the orientation of the comparison?

How much does the NR slope in Figure 5D change if you remove the outlier sample (far right point)?

Labels in Figure 6B scatter plots are illegible.

Figure 8. Please add barplots quantifying the change in intensity of HLA and CD8 – it is difficult to accurately assess by eye.

The label DHA ASTX727 is first used in the discussion.

lack/defective -> lack of/defective

There is a period missing after “each tumor sample (Supplementary Table 5)”

that the group High-ICR/GIE was -> that the HIGH-ICR/GIE group was

However, further support the notion -> However, further support for the notion

concordat genes -> concordant genes

PD: progression disease -> PD: progressive disease

Reviewer #4 (Remarks to the Author): with expertise in epigenomics, cancer

The study by Noviello et al. is interesting and addresses novel findings that could have translational value. However, it also has important limitations. My detailed assessment is provided below.

A. Abstract: this key section of the paper is not sufficiently informative, and it could be substantially enhanced. For eg. some statements are too technical, even for someone experienced in the field. GIE and ICR need to be better explained, otherwise, paraphrased into more meaningful information. On the contrary, other terms are too broad, lacking informative meaning (eg. "Some genomic features"). The purpose of mentioning survival details is not clear. There are also too many abbreviations, some of which are not abbreviated (eg. OS and DoR, even though they may be familiar terms to readers from the field).

B. Introduction: ICI needs to be first defined and supported with more contextual information before discussing its effect on prognosis.

C. Results:

1. R and NR require abbreviation in text (not just in figure legends).
2. The number of figures is unnecessarily large, some figures are too thin, and some figures have redundant titles. Hence, certain figures could be rather grouped with other figures. This is not just for formatting purposes but also to enhance the contextual information and interpretation of related findings.
3. Fig 3A: The following statements require statistical analysis to validate them: "This analysis showed a progressive enrichment from baseline to week 12 in Gene Ontology Biological Processes (GO:BP) categories related to immune processes in R compared to NR

patients”, and “In contrast, in lesions from NR patients, a progressive increase from baseline to week 12 was found for GO terms related to adhesion, cell cycle, metabolism, and skin developmental processes”.

4. Fig 4: For the statement “Several of the genes in the ICR signature showed a progressive expression increase in R patients during therapy, compared to NR patients”: the heatmap visually demonstrates that for the R patients. However, for the NR patients, this is not clearly the case for the associated statement: “By contrast, lesions from NR patients showed significantly higher expression of cell cycle-, EMT- and skin development-related genes”. Moreover, the figure legend states that “Significance values on the right correspond to the difference correspond to all weeks together (p-value from Student's t-Test, ∴ p <0.1, *: p <0.05, **: p < 0.01) (B)”. Student’s t-Test is not the appropriate test to compare more than two items, such as all weeks together.

5. Fig 5A: The following statement requires statistical analysis: “Interestingly, the specific immune subtypes that discriminated against R from NR lesions also showed a progressive increase over time of treatment in R subjects.”

6. Fig 6: It is not clear which data/figure supports the following statement “A rank aggregation analysis of gene expression and gene methylation in NIBIT-M4 tumor biopsies demonstrated the interdependent changes between gene body methylation levels and gene expression induced by guadecitabine”. The statement also mentions “gene body” methylation while the subsequent statements focus on gene promoters; is there a missing link here? Also, in panels B, the axes indicate $-\log(p\text{values})$ while the pertinent text actually refers to effect sizes (hyper/hypo methylation and up/down regulation) and not to p value significance. It seems the text of the results are not well aligned with the figures or that there are some errors in the labels of the figures, etc...

7. The last Result section is relatively lengthy. It could benefit from further focus. For eg. some of its statements may better fit in the Discussion, such as “In fact, lesions with defective expression of HLA class I molecules may retain evidence for development of adaptive immunity (High-ICR) and also for CD8+ T cell infiltration, but downmodulation of MHC class I molecules on tumor cells prevents recognition of HLA/neoantigen complexes by T cells, thus suppressing the possibility of immunoediting (therefore the lesions are identified as Non-GIE)”. Also, the part pertinent to Fig 9 may be worth putting as a separate section, focusing on replication of findings.

D. Methods

1. Survival analysis does not address adjustment to important confounders such as patient age, tumor stage, therapy type, etc...
2. A section on statistical power is necessary.

E. Discussion:

1. BOR requires abbreviation.
2. Some statements may need to be toned down. For eg, the following statement is based on only few rare events: "These two novel findings represent a first selection strategy to increase the success of similar studies". Another statement is "One of the most interesting messages of our data is that the most evident difference between patients is the dynamic increase of the level of NK-cells and CD8 T-cells in patients that respond to therapy, rather than the tumor microenvironment composition at the baseline." I may have missed something here, but the role of the tumor microenvironment cannot be totally ruled out (esp. given the limited sample sizes used for this analysis).

Overall:

1. Robustness of the findings: sample size is very limited (as also acknowledged by the authors) especially relative to the high-dimension omics analyses; this drastically limits statistical power. Increasing samples sizes and/or replication in similar trials is crucial. Replication was nicely performed for a portion (but not all) of the results (in Fig 9). A methods section on statistical power is missing. These points need to be addressed collectively.
2. UV exposure as a missing link: it seems reasonable to address UV exposure in this study (which seems to be a missing link that can markedly improve the manuscript) especially because: (1) melanoma is largely driven by exposure to UV radiation, (2) the analysed samples already have genetic data that can allow the inference of UV exposure through UV mutational signatures, (3) there have been recent studies highlighting the impact of UV exposure on DNA methylation alterations, mutations, transcriptional regulation and patient prognosis based on melanoma tissues (all of which are data available in this study and which could be directly exploited, and such published studies can be a useful guide for this), and

(4) UV exposure is known to associate with TMB and neoantigen load, the latter being one of the interesting aspects addressed in the study. Hence, including UV analysis can link several aspects of this study to one another and enhance the overall interpretation and impact of the study.

3. Coherence: the manuscript may benefit from further aligning and focusing of several aspects in relation to one another (abstract, structure of figures, Results, etc...). Some detailed ways to enhance this was provided in my above comments in pertinent sections.

Reviewer #5 (Remarks to the Author): with expertise in immuno-epigenetics, clinical trials

Overall message:

The authors present a 5 year follow up of the NIBIT-M4 study combining ipilimumab + Guadecitabine in newly diagnosed stage III and IV previously untreated melanoma patients. They previously published results from this study in Clin. Cancer Res. 25, 7351–7362 (2019). They now update their survival data and provide extensive multi-omic correlative analysis performed on serial biopsies from 14 of the 19 patients treated on study. Responses were observed in 6 patients with 3 of these having long term response off treatment and 3 additional patients requiring salvage therapy but with prolonged survival/duration of response. Serial samples are available from all the responders.

The authors have performed an enormous amount of work on these samples. Using multi-omic analysis approaches they have examined changes in global gene expression by whole exome and RNA sequencing as well as methylation using reduced representation bisulfite sequencing. They identified differences in gene signature between responding and non-responding patients to this combined immunotherapy (CTLA4 blockade) in combination with an experimental hypomethylating agent, guadecitabine.

They suggest that the best responses to their therapy were seen in individuals demonstrating both baseline evidence for immune activation and tumor recognition (as characterized by evidence in the tumor of a signature associated with FEWER mutations

compared to the number predicted) what they call the genetic immunoeediting (GIE) score in combination with the immune constant of rejection score, an established score which describes the number of immune active cells in a tumor environment, thus the combined score provides a sense of both # and function of immune cells within the tested tumors.

Overall the paper is of significant interest and the authors have produced a wealth of data.

Unfortunately, it is somewhat hard to follow the thread of their story and their takeaway points remain somewhat poorly defined. I would very much like to accept this paper as I think it adds substantially to the field, but some careful editing and more clarity in presentation of the data, assessment of what they have found and in summary of what we should do with their results would be helpful. I have attached a word document with some specific minor suggestions for clarity, but these do not address the main problem with this paper, which is a lack of clear message.

This issue could be readily addressed by edits to the discussion and results sections to provide the reader with a summary of what they have found in each figure and how they put these data together for the field.

Minor Comments

This paper uses excessive acronyms many of which are similar (ICI, ICR, etc) which make readership challenging for the non-cognoscenti.

Page 5 line 109: please define DCR.

Consider changing the acronym used for DNA hypomethylating agents from DHA to either DNA methyltransferase inhibitors (DNMTi) or hypomethylating agents (HMAs) these abbreviations are perhaps used more commonly within the field and might be easier to follow. As the authors are no doubt aware, these drugs hypomethylate RNA as well as DNA and also cause DNA damage.

Responding (R) and non-responding (NR) patients are abbreviated as indicated here. In figure 6 R is also used to indicate spearman correlation coefficients rho. I would suggest not conflating these two abbreviations. The R and NR are defined in the figure legends, but not in the manuscript text the first time these abbreviations come up on page 5 line 112.

Queries:

Page 13 lines 357-361: The authors discuss that their work identified specific melanomas which appear to have downregulated or developed defects in antigen processing and presentation characterized by loss of MHC class 1. These authors and others in the field have demonstrated that malignant cells treated with DNA methyltransferase inhibitors can upregulate expression of MHC proteins. They do not clearly discuss this in the results section. One might hypothesize based upon these prior data that the DNA methyltransferase inhibitors might improve the response to immune checkpoint therapy most dramatically in melanomas with high ICR scores but lower evidence of immune editing, and moreover, serial monitoring of these relative scores might show increased expression of MHC class 1 in the DNMTi treated samples. ADDENDUM: Figure 4A page 24 line 607 seems to suggest that among responders there was upregulation in HLA class 1 and 2 expression, while non-responders did not seem to demonstrate this change as clearly, particularly for class 2 expression which appears to be uniformly down in the non-responders. Please comment on this in the results and discussion section as this seems to be a very important finding.

Results page 5 lines 104-110 and Page 14: the original study NIBIT-M4 enrolled pre-treated or untreated patients with melanoma to receive guadecitabine + ipilimumab. They had 6 long term survivors with 3 long term CRs who were off study and 5 individuals with durable CRs. Among the 3 who recurred but were not alive, what were the subsequent lines of therapy? Did any of these patients receive 2nd generation ICI therapy and did they have a better than expected response? (I see these data in the figure legend, but it is not discussed in the manuscript text, I would briefly mention it there) and it would be nice to delineate which of the patients received which therapy on the swimmer's plot.

Page 14: you have RRBS on all the serial samples. What did this show? In the tumors with mutation in DNMT1 and SETD2 show hypomethylation of the tumors at baseline without

evidence of an effect from the DNMTi to support the authors hypothesis they suggest for resistance? These are single patient series, but might be hypothesis generating.

Page 23 Figure 3A: I presume that the responders in 3A are in peach/orange and the non-responders are in green as listed in figure 3B but I would like this to be spelled out in the figure legend please. I don't really understand or follow what is being presented in this figure 3A. What does normalized enrichment score mean? Does the spread from baseline to on treatment indicated increased transcriptional activity? I find this figure very difficult to follow.

Page 25 figure 5: I "Immune Context" might be clearer. Contexture is defined by Merriam Webster as the act, process, or manner of weaving parts into a whole or a structure so formed. The use of this description is not incorrect, but it is an unfamiliar word and I'm not sure that the authors have completely finished their 'weaving'.

Figure 9: I don't think the authors looked independently at 3 different cohorts. I believe they mean to say that they examined ICR/GIE scores from 83 patients compiled from 3 separate studies and considered them as a group against which to validate their score. This needs to be clarified in the figure legend, it is described correctly in the text.

Reply to Reviewer's comments of the paper

**Guadecitabine plus ipilimumab in unresectable melanoma:
five-year follow-up and correlation with integrated, multi-
omic analysis in the NIBIT-M4 trial**

Noviello et al.

We thank the five Reviewers for taking the time to critique our manuscript so carefully. We are extremely pleased with their comments and their very useful and constructive criticisms and suggestions. With this revision, we have taken careful steps to address the concerns and incorporate their suggestions into the new manuscript.

Here is a point-by-point reply to the reviewers' suggestions:

REVIEWER COMMENTS

Reviewer #1 (Remarks to the Author): with expertise in in bioinformatics, cancer immunology

Major comments

(1) "Genomics landscape: mutational profile differences in baseline and on-treatment lesions in R vs. NR patients" In this section, the authors observed a lot of differences. But the authors fail to link these differences in R vs NR to the mechanism of response/resistance, for DHA alone or ICI alone, or for their synergistic effects. Besides, the somatic mutations being consistent across tumor biopsies of the same patients across different time points suggest a lack of tumor evolution during treatment. This means the ¹DHA and/or ICI treatment is not very effective? If not, what does it mean? The authors need to interpret what they observed.

R: We thank the reviewer for his interesting comment. Indeed, we underline that the paper presents a retrospective analysis of the NIBIT-M4 trial with a five-year follow-up. This phase 1 trial is based on a strong rationale that hypomethylating agents can reactivate an innate immune response inducing viral mimicry ¹. Similar efforts are being pursued by other groups, as also reported in the paper. The important question posed by the reviewer about the discrimination of the role of individual treatments versus their synergistic effect is the long-term goal of our research line. It is being pursued with several forthcoming trials, such as the NIBIT-ML1 (NCT04250246). The NIBIT-M4 that we are reporting in this study, has just the combined arm. Therefore, the synergistic effect of the combination can be inferred with integrative analysis, in the absence of the arm of the single therapy. Nevertheless, the remarkable survival rate at 5yrs of 29% is far above the current state of the field, even for a limited number of patients. Therefore, we are further exploring the effect of the combination in the NIBIT-ML1 study, which is currently in the enrollment phase. Regarding the specific question of consistency of mutation across tumor biopsies, our belief is that the limited time of 12 weeks between pre- and post-therapy lesion is not sufficient to highlight resistance mechanisms. The consistency is an indication that the somatic calls are correct. However, in order to better answer the question about consistency between week, we classified the genes reported in the oncoplot of Figure 2 in tiers as reported in COSMIC database. As it is reported in **Figure R1**

below the driver genes (tier 1), we observe a decrease in variant allele frequency (VAF) in many driver genes, which suggests a diminishing of tumor content. We also point out that the decrease of the VAF is also a consequence of the increased immune infiltration and hence a lower purity in the responding patients.

Figure R1. Variant Allele Frequency (VAF) of the somatic mutations in Figure 2, classified according to the clinical relevance.

Regarding the interpretation of the results, in the revised version, we provide an improved presentation and discussion of these data. First, we are aware that it is not possible to completely disentangle the effects of the demethylating agent from those of ipilimumab in the overall mechanism of response/resistance, as this would need a specific trial in which patients are randomized to receive guadecitabine or guadecitabine plus ipilimumab. Therefore, even the impact of the genomic landscape is difficult to be linked to the DHA rather than to ipilimumab. Nevertheless, our results confirm and extend evidence of resistance obtained in ICB-only trials. First, we found that BRAF mutations are more frequent in NR compared to R patients and the opposite was true for NRAS mutations. Concerning NRAS mutations, available evidence indicates that patients with NRAS mutation-positive melanomas have improved PFS after ICB with anti-CTLA-4 and anti-PD-1 antibodies² and improved OS when treated with ipilimumab³. Concerning BRAF mutations, there are conflicting reports in the literature⁴. Some meta-analyses did not evidence differences in response between BRAF wild type and BRAF mutated patients⁵ Other studies reported a negative association between BRAF status and OS after ipilimumab in melanoma⁶. Thus, the associations that we found with BRAF and NRAS mutational status, with the caveat of the limited sample size in our trial, appear consistent with evidence obtained in clinical trials of ICB-only. These data thus suggest that the associations of BRAF/NRAS mutations with clinical response may reflect their impact on the immune checkpoint inhibitor, rather than their relevance for the mechanism of action of the demethylating agent. However, further trials are needed to settle this issue. Concerning CDKN2A alterations, they have been previously found associated with reduced benefit from ICB in urothelial carcinoma, but not in melanoma⁷. Immunotherapy resistance of CDKN2A deficient tumors has also been reported in pan-cancer studies⁸. We found the CDKN2A mutations to be more frequent in NR patients.

EPHA7 mutations, that we found more frequently in R patients, have been associated with better clinical outcomes in ICB-treated patients across multiple cancer types⁹. PLXNA4 mutations, more frequent in R patients, have been shown to promote Cytotoxic T-lymphocyte infiltration in pre-clinical tumor models, as PLXNA4 behaves as a negative checkpoint regulating T cell migration and proliferation¹⁰. In summary, available evidence indicates that most differentially enriched mutations between R and NR groups have also been associated with response or resistance to ICB-only regimens. In other words, the genomic landscape of the lesions may contribute to explaining the efficacy or lack of efficacy of the ipilimumab arm of the trial. Instead, the transcriptional landscape can tell us how susceptibility to immunomodulation by guadecitabine is crucial for the response (see replies to the following questions). This interpretation has a notable exception: the DNMT1 mutations found in two NR patients. DNMT1, a guadecitabine target, is the DNA methylation “maintenance” enzyme at the time of cell division, and finding it mutated in NR patients corroborates the hypothesis that resistance to guadecitabine contributes to the observed lack of clinical benefit.

(2) “Transcriptional landscape of baseline and on-treatment tumor lesions: distinct and evolving transcriptional programs distinguish R from NR patients” In this section the authors talked a lot about the differences in gene expression in the immune angle, which makes sense. But the authors didn’t mention anything about DHA. Does the difference in gene expression between R and NR reflect differential response to DHA (and maybe in combination with ICI)?

R: Reply. We thank this reviewer for asking us to address this issue. Concerning the statement “But the authors didn’t mention anything about DHA” we would like to provide an improved explanation of the results in the mentioned section of the manuscript. As shown in Figures 3B of the revised manuscript we report the expression of guadecitabine-specific gene signature that we recently defined in melanoma cells¹¹ by Whole Transcriptome profiling and by Nanostring profiling. The results indicated improved expression of this signature in on-treatment lesions from R patients compared to NR patients. This result suggests that clinical benefit requires susceptibility to the immunomodulatory action of this DNMT inhibitor. A consequence of this hypothesis is that NR patients must have some resistance mechanism to guadecitabine, and the DNMT1 mutations found in two NR patients may be part of this resistance mechanism. However, to gain further insight into the transcriptional effects that could be ascribed to guadecitabine, and that could contribute to explain clinical benefit in this combinatorial trial, we took advantage of recent evidence indicating the relevance of tumor-specific antigens derived from transposable elements¹² and ERV sequences¹³ for the immune response to tumors, even in the context of ICB. Both these classes of sequences are known to be regulated by methylation and to be susceptible to re-activation by demethylating agents. Specifically, we tested the hypothesis that clinical benefit in the NIBIT-M4 trial could depend on the ability of guadecitabine to re-activate these sequences, by promoting their demethylation, leading to enhanced expression in R patients compared to NR patients. By inducing expression of ERV sequences, demethylating agents can activate the viral mimicry response secondary to intracellular recognition of viral dsRNA^{14,15}. This response explains the promotion of type I IFN and innate immunity pathways that characterize the guadecitabine-specific gene signature recently defined by us in melanoma cells¹¹. To this end, we first evaluated the overall methylation pattern across all genomic regions and found a trend indicating reduced methylation for both responders and non-responders (**Figure R2A** and Supplementary Figure S7A of the revised manuscript). However, by focusing on specific genomic regions (exons and intergenic regions) we noticed a stronger reduction in methylation at weeks 4 and 12 in R vs NR patients. A similar difference emerged for promoters, UTRs regions and other regulatory regions such as enhancers and super-enhancers. We then evaluated the specific methylation

pattern in pre vs on-treatment tumor biopsies during therapy in Long interspersed nuclear elements (LINEs), Short interspersed nuclear elements (SINEs) and long terminal repeat (LTR) including endogenous retroviral elements (ERVs). Again, we noticed that the effect of DHA was a general decreasing trend of methylation for both R and NR patients (**Figure R2B** and Figure 5A of the revised manuscript). We then correlated the expression and methylation of different LINE, SINE and LTR elements. The number of unmapped reads mapped on these elements was used as a proxy of their expression. Strikingly, this analysis showed a significant anti-correlation between methylation and expression (i.e., reduced methylation associated with increased expression) but only in R patients, whereas for NR patients we did not observe a similar inverse trend (**Figure R3** and Figure 5B of the revised manuscript). Overall, our results show that treatment with DHA induces the demethylation of genomic regions that contain transposable elements and ERVs. The effect of these epigenetic changes is associated with their higher expression in R. On the basis of these results, we hypothesize that clinical benefit in this NIBIT-M4 trial is associated with a demethylation/re-expression process involving transposable elements and ERV and occurring selectively in R and not in NR patients. This hypothesis has a corollary: that resistance to guadecitabine+ipilimumab may in fact depend also on resistance to guadecitabine. This mechanistic hypothesis is in agreement with additional evidence in our manuscript: first, as already mentioned, by nanostring analysis we found a progressive increase in expression of guadecitabine-specific signature genes in R vs NR. This signature is characterized by genes belonging to TLR, NF- κ B and type I-III IFN pathways and its enhanced expression in R patients is in agreement with the viral mimicry process activated by DNMT inhibitors. Second, again in the nanostring analysis (Figure S4B of the revised manuscript), we found promotion in R patients of several signatures associated with B cell response and with Tertiary Lymphoid Structures (TLS) formation. Interestingly, the B cell-mediated adaptive response detected in TLS of patients with NSCLC is directed to envelope glycoproteins coded for by retroviral sequences¹³. Thus, it is possible that in R patients the reactivation of ERV sequences has led to promotion of B cell-mediated responses that contribute to clinical benefit. Overall, our results show that treatment with DHA induces the demethylation of genomic regions, in particular transposable elements and ERVs. The effect of these epigenetic changes is associated with their higher expression in R patients.

A

B

Figure R2. Variation of the overall methylation pattern in various genomic regions and regulatory regions as function of time between R and NR patients (A). Variation of the overall methylation pattern in Long interspersed nuclear elements (LINEs), Short interspersed nuclear elements (SINEs) and Long terminal repeat (LTR) endogenous retroviral elements (ERV) (B).

Figure R3. Correlation between expression and methylation in LINE, SINE and LTR elements for R and NR patients.

(3) “By a custom-designed NanoString assay we then explored differential expression in R vs NR lesions of ...” I don’t understand why the authors need to resort to NanoString here. Can’t the RNA-seq data also provide this same information? Or should the authors do the same analyses for both nanostring and RNA-seq and check if concordant results show up?

R: As explained in the answer to the previous question, the nanostring platform was essential to better characterize a set of immune phenotypes represented as gene sets that otherwise would be difficult to analyze.

(4) “Integrative analysis of methylation and transcriptomic profiles during treatment” The authors did little analyses with respect to methylation and spent very little time to explain their analyses. But this is extremely important. With the current form of the manuscript, I cannot say I believe in the author’s analyses in this part. “A rank aggregation analysis of gene expression and gene methylation in NIBIT-M4 tumor biopsies demonstrated the interdependent changes between gene body methylation levels and gene expression induced by guadecitabine” What part of what figure are you talking about? Need a lot more details about how you analyzed your data. Also, according to this description, your figure and figure legend, you intersected the differences observed in methylation data and RNA expression data, and just ran some GO analyses. This doesn’t show “interdependent changes” at all. This statement is wrong. It is not like you are investigating the correlation between methylation and RNA expression

R: We agree with the reviewer that the referenced paragraph could be improved. The analysis of the whole methylation profile and the focus on the repeated element presented above are included in the manuscript's new version replacing the paragraph mentioned by the reviewer. These further analyses are aimed at uncovering the specific effect of the DHA and should clarify his concerns. We have completely rephrased the integrative analysis of methylation and transcription in both the results section and the methods section. Overall our results presented in Figures R2 and R3, and reported in the paper (Figure 5 and Figure S6), show that treatment with DHA induces the demethylation of genomic regions, in particular transposable elements and endogenous retroviruses. The effect of these epigenetic changes is associated

with their higher expression in responders. Moreover, the intersection of differentially expressed and methylated genes shows the epigenetic activation of immune functions in responders.

b) “By exploiting further the expected inverse relationship between gene expression and promoter methylation, we then identified modules of hypomethylated, immune-related genes up-regulated during treatment in R vs NR patients (shown in the upper right quadrant in each plot of Figure 6B).” I am lost here. How did you “exploit”? How did you identify the modules? The exact analytical procedure?

R: We have rephrased this point which generated confusion. In particular, the figure mentioned by the reviewer has been moved to the supplementary material as Figure S3B we have clarified that:

“We performed differential promoter methylation and gene expression analysis between R and NR (Figure S3B). We used SMITE¹⁶ to rank the upregulated and hypo-methylated genes as well as down-regulated and hyper-methylated genes between R and NR patients at different time points. Consistently with the previous observations, functional enrichment of hypomethylated and upregulated genes between R and NR resulted in biological processes associated with immune functions at week 12. Genes enriching immune response included granzymes (GZMM), interleukins and interleukin receptors (IL12, IL32, IL15, IL12RB1), cytokines and cytokine receptors (TNF, CXCR4, TNFRSF1B) and HLA genes and transactivators (CIITA, HLA-DOA, HLA-DMB, HLA-E).

(5) “By stratifying patients according to the ICR/GIE classification of week 12 biopsies we found a significant difference (p-value 0.035) in overall survival (OS) and in PFS between the High ICR/GIE and the High ICR/Non-GIE group (Figure 7C and Supplementary Figure 4B, left hand panel).” The number of patients here is very small. I am not sure how reliable the p-value is, even if it is significant. Also the presentation of Fig 7c is questionable. The censored patients need to be shown with “+” on the curves. None of the patients were censored?

R: We wholly agree with the reviewer that the low number of patients limit the robustness of the conclusions. We have also updated the figures showing the censored cases (just two cases). Nevertheless, we reported the p-value mentioning in the text that these results should be taken cautiously. However, the validation with other datasets, as reported in Figure 9, is comforting.

(6) “To validate the ICR/GIE stratification, we assembled a cohort of 83 melanoma cases treated with ICI, either anti-CTLA4 or -PD1, from previous published studies^{4,39,40} for which TMB, neoantigen load and gene expression were available” I am not sure how helpful this analysis is. These patients were treated with ICI only, but the authors’ own patients were treated by ICI and DHA. In fact, this paper’s biggest selling point is the combination therapy and now the authors seem to be backpadding on their own narrative.

R: The validation was performed on ICI only datasets. We would like to mention again that the combination ICI+DHA is the subject of a limited number of trials that have already shown promising efficacy. The NIBIT-M4 is one of those with the two recently reported trials^{17,18}. We have developed the GIE/ICR score as a biomarker of response to ICI, and we are applying it to several domains. We tried several

biomarkers of response, as reported in Figure 3 of the paper, but the only predictive biomarker in our cohort was the ICR/GIE stratification. For this reason, we believe that it is worth reporting that this stratification is a valuable tool to predict response in immune-based therapies, either mono-therapies or in combination.

(7) "This analysis confirmed the stratification of patients into 4 subsets as seen in the NIBIT-M4 cohort (Figure 9A)." This statement is simply wrong. This figure didn't confirm the existence of 4 subsets in the patients, as there is no clear bimodal distribution on the GIE axis, or on the ICR axis. The 4 subsets the authors identified were forced unnaturally.

R: Many thanks for raising this point that allows us to better clarify how we segregate the samples into the four groups GIE/High-ICR non-GIE/ICR, GIE/low-ICR, and no-GIE/low-ICR. These categories are not obtained by unsupervised clustering but are selected according to the values of the GIE ratio and the ICR enrichment. In particular, since the GIE is a ratio between observed and expected neo-antigens, we use $GIE < 1$ or $GIE > 1$ to nominate GIE or non-GIE samples. Analogously, we adopted the normalized enrichment score (NES) to evaluate the activation of the ICR at a single-sample level using our yaGST tool¹⁹. A value of $NES > 0$ means positive activation, whereas a value of $NES < 0$ means a significant negative activation. Therefore, we use this value to nominate high-ICR versus low-ICR. Due to the low number of samples, we found this option to be the most natural, robust, and non-arbitrary choice. The prognostic features of the GIE/ICR stratification, with the same thresholds, has been recently validated also in a large cohort of Colon cancer²⁰.

(8) Overall, the genomics analyses in this paper rely too much on gene ontology analyses. The authors need to perform different types of genomics analyses from different angles, to cross validate the same story. Ideally, some orthogonal experiments should be performed to validate key elements of their story. At the very least, GO analyses cannot distinguish the direction of gene regulation. For example, a GO term could have genes that are supposed to be up- or down-regulated in a certain condition. The authors may input a list of genes that are all up-regulated, which show a big overlap with this GO term. But those genes that are supposed to be down-regulated are in fact up-regulated, making this GO term a false positive. Many of the authors' figures show only simple GO analyses, without any other orthogonal analyses to rule out these issues, making them questionable.

R: To better answer the concerns of the reviewer, we have performed different types of investigations on our data. First, we considered the set of genes specifically induced by the guadecitabine in melanoma that we have recently described¹¹. We performed a comparative profiling of gene signatures induced by several epigenetic drugs in melanoma cell lines. We found that guadecitabine activates several immune pathways by upregulating a signature of 166 genes. Therefore, to better disentangle the effect of the DHA in our cohort, we performed gene set enrichment analysis of the guadecitabine-specific signature on the list of differentially expressed genes between R and NR (**Figure R4A**, and Figure 3B of the revised manuscript). We found that this signature is significantly activated in R at week 4 (p-value 4.17e-5) and week 12 (p-value 1.0e-4). To further validate this finding, we used the data from another trial of combining ICI and epigenetic drug in ovarian cancer¹⁷. We observed a significant activation of the guadecitabine signature after treatment (**Figure R4B** and Figure 3C of the revised manuscript). These results confirm the immune-modulatory effect of DHA *in vivo* and that this effect is associated with response.

Figure R4. GSEA enrichment of the guadecitabine-specific signature on the ranked list of differentially expressed gene at week 4 and week 12 in NIBIT-M4 trial (A) and combined therapy ICI plus HMA from Chen et al. 2022 ¹⁷(B).

Second, to further address the suggestion to carry out further orthogonal analysis taking into account the direct effect (positive or negative) of a gene on specific pathways, we exploited Ingenuity Pathway Analysis (IPA) to identify canonical pathways differentially regulated in R vs NR patients during treatment and to provide some direct visualization of the direction of gene regulation that is lost with the GO analyses. Indeed, IPA database accounts for positive and negative associations of genes and pathways.

This analysis predicted significant activation in R patients of several immune-related pathways including the “Pathogen induced cytokine storm”, the “TH1” and “TH2”, the “phagosome formation” and the “cross-talk between dendritic cells and NK cells” pathways (Supplementary Figure S4C of the revised manuscript). In contrast pathways predicted to be inhibited in R vs NR patients included the “PD-1, PD-L1 cancer immunotherapy”, the “MSP-ROn signaling in macrophages” and the “GP6 signaling” pathways. The actual direction of regulation of significant pathways in R vs NR patients was visualized in Supplementary Figure S4C of the revised manuscript. These results confirm that R patients experience the activation of immune pathways crucial for developing innate and adaptive immunity.

Minor comments

(1) *“Previous studies have shown that genetic evidence of neoantigen depletion is preferentially observed in lesions with high Immunoscore”. Need to explain what is “immunoscore” and add a citation for it. “while progressing clones tend to be immune privileged” Same for “immune privileged”*

R: We referred to the paper of Angelova et al (Cell 2018) ²¹ showing the immunological features of the longitudinal reconstruction of clones in 31 metastases. These immunological features were measured using the Immunoscore ²² and quantification of immuno-editing ²³.

(2) *“The presence of an adaptive immune response within tumors is accounted for by the Immunological Constant of Rejection (ICR) 19,20.” Need one sentence in this paragraph, immediately following this sentence, to explain what ICR really measures and what it means*

R: Thank you for the suggestion we have specified that:

“The ICR signature incorporates interferon-stimulated genes driven by transcription factors *IRF1* and *STAT1*, with *CCR5* and *CXCR3* ligands, immune effector molecules and counter-activated immune regulatory genes. A high expression of ICR genes typifies ‘hot’/immune active tumors characterized by the presence of a T helper 1 (Th-1)/cytotoxic immune response and predicts survival and response to ICI in different tumors including Colon, Breast, Bladder, Stomach, Head and Neck, Sarcoma, and Melanoma.”

(3) *A major drawback of this study is the small number of patients. But I understand there is not too much the authors can do about this and the current collection of data is already a significant effort. But I feel it might still be better to acknowledge this deficiency, especially in places where sample size is critical. This includes, for example, the section on tumor mutations (Fig. 2). The data are very sparse in the waterfall plot, and the “enrichment” of genes is really weak*

R: We fully agree with the referee. We are aware of the limited number of samples, and the extensive analysis reported in the paper aims to learn as much as possible from this unique dataset. We have extensively underlined that the enrichments here should be taken with caution and reported this statement in the parts describing the limits of the study of the Discussion.

Reviewer #2 (Remarks to the Author): with expertise in melanoma, clinical

This manuscript focuses on the clinical and translational results from an Italian cooperative group study utilizing ipilimumab with the demethylating agent guadecitabine in advanced melanoma patients. It is hypothesized that demethylating agents will work well with ICI as these agents are known have immunomodulatory effects on tumor cells and activate innate immune pathways. There is preclinical evidence supporting the combination therapy as well showing guadecitabine is able to promote gene expression of interferons and TLRs (known to be implicated in innate immune response).

The plan for this trial was to perform longitudinal multi-omics profiling including WES, RNAseq and reduced-representation bisulfite sequencing on tumor biopsies collected at baseline, week 4 and week 12 on 14 of the treated patients.

Key findings include:

The clinical outcome data was impressive for an ipi-based regimen with 5 year OS at 28.9% which is higher than the traditional 20% seen on single-agent ipi trials.

No difference in TMB in R vs NR. NRAS more common in R, BRAF more common in NR. EMT genes more common in NR. CDKN2A more in NR. Dynamic increase in level of CD8 and NK cells in responders was more important than the baseline immune cell composition.

Ongoing randomized phase 2 study to test efficacy of addition of ASTX737 to ICI in pd1 refractory melanoma.

The bulk of the manuscript was in describing the multi-omics data with very little provided in terms of clinical data as it has been previously reported. The work was original and the conclusions were supported by the provided data. There are no clear flaws in data analysis or interpretation and methodology is sound. The investigators should be applauded for collecting so many good quality biospecimens to allow such detailed analyses.

R: We thank this reviewer for his comment. We were delighted that he/she found our work interesting. Epigenetic immuno-modulation is a suitable strategy to improve response to immune therapies and this kind of work can have an impact.

Q) Were patients PD1 refractory or naïve?

R: We thank the reviewer for this question that allows us to implement the manuscript with this information. Eighteen patients (95%) were treatment-naïve at study entry and only 1 (5%) had received PD-1 mAb as first line therapy.

Q) Could you be more specific about categorizing patients as R or NR based on "Disease Control." How was disease control defined?

R: Based on the reviewer's suggestion, we have specified how Disease Control (DC) was defined, and how patients were categorized as R or NR, in the revised version of the manuscript (Materials and Methods). Specifically, patients were classified as R if they experienced a DC [defined as Complete Response (CR), Partial Response (PR), or Stable Disease (SD)], while patients who experienced a Progressive Disease (PD) were classified as NR.

Q) Are there any further studies planned with guadecitabine in melanoma or is the focus on the ASTX compound that is being used in the randomized phase 2 study?

R: No further clinical studies are planned with guadecitabine. Its oral formulation (ASTX727) has recently been clinically available and it has demonstrated an almost identical PK/PD effect compared to i.v. decitabine. Thus, providing also a significant dosing advantage for patients as compared to s.c. guadecitabine, ASTX727 is under investigation in the randomized phase 2 study NIBIT-ML1 (NCT04250246) as correctly mentioned by the reviewer.

Reviewer #3 (Remarks to the Author): with expertise in cancer genomics, immunology

Summary: Noviello et al analyze matched pre- and on-treatment samples for the NIBIT-M4 clinical trial of unresectable melanoma treated with the combination of guadecitabine and ipilimumab. They analyze somatic variation and gene expression in responders and nonresponders across time points. Using a number of scores capturing characteristics of the tumor and tumor immune microenvironment, they find that no single score accurately predicts which patients will respond. However, a combination of two scores ICR and GIE which capture evidence of active immunity at the tumor site and immunoediting respective, is capable of predicting which patients are likely to respond. They validate this score in independent cohorts. They also note that responders tend to have increased immune signatures at later time points whereas non-responder tumors do not. Overall, there are few human studies with genomic and transcriptomic data from on-treatment samples, and the analysis presented here is likely to be of broad interest to the community. The ICR/GIE score seems promising as well. Overall the manuscript is clearly written. Certain aspects could use further clarification.

R: We appreciate the reviewer's evaluation and consider all comments for this revision. We believe that his/her comments, as well as the suggestions of the other refereed, contributed to substantially improving our paper.

Major Comments:

Q: It was not entirely clear how the authors compared across time points for specific analyses (gene expression, immune infiltration, methylation). For such analyses (e.g. Figure 3A, 5A), did the authors control for non-independence of different timepoints for the same? Can the authors quantify the effect size / significance of the increase between timepoints from in 3A? Right now, the reader is asked to accept the increase based on the visual representation, but the underlying calculation (i.e. enrichment for certain GO:BP gene sets over time) is complex so the interpretation is not entirely straightforward.

R: We acknowledge that the original Figure 3A has a difficult interpretation. We want to clarify that the analysis reported in the figure is the differential gene expression per week between R and NR, therefore we used multiple pair-wise t-tests. In the revised manuscript we avoided to pool samples from multiple time points. When appropriate, we have used two-way mixed anova, as specified in the legends.

To quantify the effect/significance of the difference between time-points, the updated figure (**Figure R5**, and Figure 3A of the revised manuscript) displays the mean of the Normalized Enrichment Score and the aggregated p-value using Fisher's method. We hope the new figure with a quantitative representation of the functional enrichments can better convey that the gene expression landscape indicated distinct and evolving transcriptional profiles characterizing baseline and on-treatment tumor biopsies from R compared to NR patients. Lesions from R patients showed progressive enrichment for signatures and gene sets revealing activation of adaptive immunity and effective immunomodulation.

Figure R5. Transcriptional Landscape of NIBIT-M4 trial. Supervised differential analysis between responders and non-responders before treatment (week0) and after four (week4) and twelve (week12) weeks. The x axis reports the aggregate p-value by Fisher's method. Enriched Biological Process Gene Ontology (GO) terms are classified into seven main categories. Size of the dot represents the number of GO terms grouped into a category; color of the dots is the mean Network Enrichment Score of the GO terms.

Q: It was unclear to me whether the use of guadecitabine could be credited for aiding the immune response in responders. The authors note that in responders, there is increasing hypomethylation of immune genes over time relative to non responders. However, there are also higher expression levels of immune genes and higher levels of immune infiltrate. How do the authors rule out that a higher proportion of DNA coming from cells actively expressing those immune genes explains the change in methylation, rather than the

drug. Did the authors note changes to methylation from baseline in general? I.e. rather than perform differential methylation between R and NR at different time points, did the authors compare differential methylation within R and NR groups over time?

R: We followed the suggestion of the reviewer and firstly evaluated the overall methylation pattern across all genomic regions and found a trend indicating reduced methylation for both responders and non-responders (**Figure R2A** and Supplementary Figure S6A of the revised manuscript). However, by focusing on specific genomic regions (exons and intergenic regions) we noticed a stronger reduction in methylation at weeks 4 and 12 in R vs. NR patients. A similar difference emerged for promoters, UTRs regions and other regulatory regions such as enhancers and super-enhancers.

However, to gain further insight into the transcriptional effects that could be ascribed to guadecitabine, and that could contribute to explain clinical benefit in this combinatorial trial, we took advantage of recent evidence indicating the relevance of tumor-specific antigens derived from transposable elements¹² and ERV sequences¹³ for the immune response to tumors, even in the context of ICB. Both these classes of sequences are known to be regulated by methylation and to be susceptible to re-activation by demethylating agents. Specifically, we tested the hypothesis that clinical benefit in the NIBIT-M4 trial could depend on the ability of guadecitabine to re-activate these sequences, by promoting their demethylation, leading to enhanced expression in R patients compared to NR patients. By inducing the expression of ERV sequences, demethylating agents can activate the viral mimicry response secondary to intracellular recognition of viral dsRNA^{14,15}. This response explains the promotion of type I IFN and innate immunity pathways that characterize the guadecitabine-specific gene signature recently defined by us in melanoma cells¹¹. We evaluated the specific methylation pattern in pre vs. on-treatment tumor biopsies in Long interspersed nuclear elements (LINEs), Short interspersed nuclear elements (SINEs) and long terminal repeat (LTR) including ERVs. Again, we noticed that the effect of DHA was a general decreasing trend of methylation for both R and NR patients (**Figure R2B** and Figure 5A of the revised manuscript). We then correlated the expression and methylation of different LINE, SINE and LTR elements. The number of unmapped reads mapped on these elements was used as a proxy of their expression. Strikingly, this analysis showed a significant anti-correlation between methylation and expression (i.e., reduced methylation associated with increased expression) but only in R patients, whereas for NR patients we did not observe a similar inverse trend (**Figure R3** and Figure 5B of the revised manuscript). Overall, our results show that treatment with DHA induces the demethylation of genomic regions that contain transposable elements and endogenous retroviruses. The effect of these epigenetic changes is associated with their higher expression in responders. Based on these results we hypothesize that clinical benefit in this NIBIT-M4 trial is associated with a demethylation/re-expression process involving transposable elements and ERV and occurring selectively in R and not in NR patients. This hypothesis has a corollary: that resistance to guadecitabine+ipilimumab may depend also on resistance to guadecitabine. This mechanistic hypothesis agrees with additional evidence in our manuscript: first, as already mentioned, by the gene expression analysis we found a progressive increase in the expression of guadecitabine-specific signature genes in R vs NR (**Figure R4** and Figure 3C and 3D of the revised manuscript). This signature is characterized by genes belonging to TLR, NF- κ B and type I-III IFN pathways and its enhanced expression in R patients agrees with the viral mimicry process activated by DNMT inhibitors. Second, again in the Nanostring analysis (Supplementary Figure S4B of the revised manuscript), we found promotion in R patients of several signatures associated with B cell response and with Tertiary Lymphoid Structures (TLS) formation. Interestingly, the B cell-mediated adaptive response detected in TLS of patients with NSCLC is directed to

envelope glycoproteins coded for by retroviral sequences¹³. Thus, it is possible that in responding patients the reactivation of ERV sequences has led to the promotion of B cell-mediated responses that contribute to clinical benefit. Overall, our results show that treatment with DHA induces the demethylation of genomic regions, particularly transposable elements, and endogenous retroviruses. The effect of these epigenetic changes is associated with their higher expression in responders.

Q: What genes were affected in non-responders? Presumably non-response to an immunotherapy does not necessarily imply non-response to a demethylating agent, so one might expect to see some effects attributable to guadecitabine in the NR that may be more reflective of changes in methylation more generally. What genes were affected in the absence of a robust immune response?

R: We agree this is another important question that could be addressed using our longitudinal data. We did not include this analysis in the previous version. The answer to the question can be obtained by independently comparing the expression profiles of R and NR in different time points. As in all previous analyses, the specific effect of the guadecitabine cannot be completely decoupled from the effect of the checkpoint inhibitor. Therefore, we used SMITE¹⁶ to rank the upregulated and hypo-methylated genes in R and NR patients between week 12 and the baselines. This analysis should uncover the pathways activated on-treatment. The top 100 genes of the ranked list were used for functional enrichment. The genes of the list were associated with fate determination (WINT5A), cell migration (PRKD2), epithelial differentiation (KRT15), and other pathways related to skin and epidermis development/differentiation together with pathways related to motility (Figure R6A and Supplementary Figure S7A of the revised manuscript). On the contrary, after treatment, the same longitudinal analysis of lesions from R patients resulted in the activation of immune response functions such as regulation of T and natural killer cell activation and proliferation (IL15), Th1 polarized T-cells (TBX21), and other functional categories associated with lymphocyte activation (Figure R6B and Supplementary Figure S7B of the revised manuscript). These results show a differential effect on the combined treatment between R and NR patients.

B
Figure R6. Heatmap displaying the relationship between enriched biological processes and genes that are simultaneously upregulated and hypomethylated at week12 vs week0 (A) in non responder group and (B) responder group. Color of the boxes represents the significance of the enrichment.

Q: Does GIE change if the authors limit to mutations that show evidence of RNA support (variant allele present in RNA reads)? One would expect that removing mutations that are not expressed would strengthen the signal if anything.

R: The neo-antigen call pipeline used in the original manuscript included the expression of the gene as a filtering step to call neo-epitopes. We updated the neo-antigen call pipeline to account for the reviewer's request. We used the number of mutant reads in the RNA-seq alignment file as a further filtering step for neo-antigen call. We select the mutations with high binding affinity with the HLA (less than 500 nM), Differential Agretopicity Index < 1 and with at least one supporting read on the RNA-seq data. The results presented in the new figures 6 and 8 of the revised manuscript are confirmed.

Q: Also, the motivation for the score cutoffs used to designate ICRhigh and GIEhigh was not entirely clear. Are the authors splitting on the median score? Since the basic cutoffs validate, could the authors try to see if learning a cutoff based on NIBIT-M4 improves separation in the validation cohort (maybe using a regression model?). Could the ICR/GIE score be combined with other covariates to improve prediction based on the pre-treatment sample?

R: Many thanks for raising this point that gives us the opportunity to better clarify how we segregate the samples into the four groups GIE/High-ICR non-GIE/ICR, GIE/low-ICR and no-GIE/low-ICR. These categories are not obtained by unsupervised clustering but are selected according to the values of the GIE ratio and the ICR enrichment. In particular, since the GIE is a ratio between observed and expected neoantigens, we use $GIE < 1$ or $GIE > 1$ to nominate GIE or non-GIE samples. Analogously, we adopted the normalized enrichment score (NES) to evaluate the activation of the ICR at single-sample level using our yaGST tool¹⁹. A value of $NES > 0$ means positive activation, whereas a value of $NES < 0$ means a significant negative activation, therefore we use this value to nominate high-ICR versus low-ICR. Due to the low number of samples, we found this option to be the most natural, robust, and non-arbitrary choice. Learning the cutoffs would require a cross-validation procedure that is unfeasible given the size of the cohort. Moreover, this could make the GIE/ICR stratification cohort-dependent and difficult to generalize. On the contrary, the natural cutoffs used to nominate the four groups and their validation in external datasets show the robustness of our approach. The prognostic features of the GIE/ICR stratification, with the same thresholds, has been recently validated also in a large cohort of Colon cancer²⁰. Regarding the covariates, we agree that this could be a viable idea for improving the prediction. However due to the limited sample size we avoid to over-parametrize our analysis.

Q: Did the authors note a loss of DNA/RNA support for neoantigens over time in responder lesions relative to non-responders? This analysis may need to control for tumor content of the samples though, as immune cells will be more abundant in on-treatment samples, potentially dominating the DNA / RNA content of the sample.

R: This is an intriguing question, as we report in the answer to the question that follows, we were able to estimate the purity just using the RNA-sequencing data. The data shows, as expected, a lower tumor content and higher immune infiltration over time in R. However, this does not result in real differences between the allele frequency and/or clonality of the neoantigens. We plan to explore this in future studies with more samples.

Q: The authors conclude from expression analysis that immune surveillance suppresses the cell cycle and differentiation pathways. This is one possibility. Did the authors evaluate whether the apparent reduction in these pathways could reflect that there are many more immune cells contributing RNA to the sample and thus it could be a relative effect? Could this be normalized to the tumor cell content of the samples?

R: The reviewer is raising an important point about quantifying the specific contribution of malignant versus non-malignant cells when performing differential pathways analysis. There are DNA-based several state-of-the-art tools such as Absolute²⁰ and others that require accurate copy number variation (CNV) calls. In our cohort, for about half the patients we lack the exome sequencing of the reference normal blood. Therefore, we are not completely confident of the CNV calls for these samples. Indeed, the oncoplot of Figure 2 reports just the confident CNV calls. We tried to quantify the purity from RNA-seq using a validated method such as ESTIMATE²¹, which uses a combined approximate quantification of the

immune and stromal score. We report in **Figure R6** below, the variation of Estimate score between week 0 and week 12 in R and NR patients. This analysis shows that in four out of six R patients, the Estimate score increases, versus three out of seven NR cases with an increasing Estimate score. This confirms, as suggested by the reviewer, that R patients tend to have an increased number of immune cells and a lower purity. However, for an accurate answer to this specific question, the ideal platform would be the use of single-cell RNA sequencing. This is one of the topics we are exploring further in the next studies, and we prefer to not report this partial result as it can be misleading due to the low number of patients and the lack of an accurate purity estimation.

Figure R7. Line plot showing for each patient the variation of the ESTIMATE score between week 12 and week 0 in non responder (left) and responder (right) group.

Q: One of the melanoma differentiation scores, MPS, was suggested to relate to mechanism of immune evasion. It might be interesting to look at MPS score as well: Pérez-Guijarro, E., Yang, H.H., Araya, R.E. et al. Multimodel preclinical platform predicts clinical response of melanoma to immunotherapy. Nat Med 26, 781–791 (2020). <https://doi.org/10.1038/s41591-020-0818-3>

R: As suggested by the reviewer, we decided to include the MPS as another marker of immune evasion. It is considered in Figure S4A of the revised manuscript together with other biomarkers.

Q: It was not clear whether multiple testing correction was performed or what methods was used to do so.

R: We have amended all the figure legends specifying the used test and whether we applied multiple test correction.

Minor Comments:

Q: Consider using lower case r to denote pearson's correlation coefficient to avoid confusing with R indicating the responders.

R: Thank you for checking. We used r for correlation in the figure.

Q: Why are the $-\log_{10}p$ values in the bottom half of 3A negative but positive in the top half? Shouldn't p -value be independent of the orientation of the comparison?

R: We have completely re-drawn Figure 3. The current version has been amended following the concerns of this and other reviewers.

Q: How much does the NR slope in Figure 5D change if you remove the outlier sample (far right point)?

R: We have replaced the analysis reported in Figure 5D with robust interpolation. The new figure confirms the correlation between NK abundance and TCRB clonality.

Q: Labels in Figure 6B scatter plots are illegible.

R: The Figure 6B has been moved to the Supplementary as Figure S6B and the size of the labels has been increased.

Q: Figure 8. Please add barplots quantifying the change in intensity of HLA and CD8 – it is difficult to accurately assess by eye.

R: We thank the reviewer for the suggestion, Figure 8 has been modified accordingly.

Q: The label DHA ASTX727 is first used in the discussion.

Q: lack/defective -> lack of/defective

Q: There is a period missing after "each tumor sample (Supplementary Table 5)"

Q: that the group High-ICR/GIE was -> that the HIGH-ICR/GIE group was

Q: However, further support the notion -> However, further support for the notion

Q: concordat genes -> concordant genes

Q: PD: progression disease -> PD: progressive disease

R: Many thanks for the accurate checking, we have corrected all the above and accurately reviewed the whole document.

Reviewer #4 (Remarks to the Author): with expertise in epigenomics, cancer

The study by Noviello et al. is interesting and addresses novel findings that could have translational value. However, it also has important limitations. My detailed assessment is provided below.

A. Abstract: this key section of the paper is not sufficiently informative, and it could be substantially enhanced. For eg. some statements are too technical, even for someone experienced in the field. GIE and ICR need to be better explained, otherwise, paraphrased into more meaningful information. On the contrary, other terms are too broad, lacking informative meaning (eg. "Some genomic features"). The purpose of mentioning survival details is not clear. There are also too many abbreviations, some of which are not abbreviated (eg. OS and DoR, even though they may be familiar terms to readers from the field).

R: The suggestion is well received. We revised the abstract according to the reviewer's suggestion. Avoiding acronyms with a more specific summary. The Journal's constraint of 150 words for the abstract imposes to be particularly brief and aimed at a general audience.

B. Introduction: ICI needs to be first defined and supported with more contextual information before discussing its effect on prognosis.

R: We have briefly introduced ICI in the first paragraph of the introduction referring to the review of Sharma and Allison²².

C. Results:

1. R and NR require abbreviation in text (not just in figure legends).

R: In agreement with the reviewer's suggestion, we abbreviated R and NR also in the text of revised version of the manuscript.

2. The number of figures is unnecessarily large, some figures are too thin, and some figures have redundant titles. Hence, certain figures could be rather grouped with other figures. This is not just for formatting purposes but also to enhance the contextual information and interpretation of related findings.

R: We have tried to follow the reviewer's suggestion, together with the suggestion of other reviewers to perform further analyses. We have eliminated one figure (the previous Figure 4), as the same message has been conveyed in other figures. We have amended several other figures in the paper.

3. Fig 3A: The following statements require statistical analysis to validate them: "This analysis showed a progressive enrichment from baseline to week 12 in Gene Ontology Biological Processes (GO:BP) categories related to immune processes in R compared to NR patients", and "In contrast, in lesions from NR patients, a progressive increase from baseline to week 12 was found for GO terms related to adhesion, cell cycle, metabolism, and skin developmental processes".

R: The figure 3A has been completely redrawn according to the suggestions. We quantified the aggregated p-value (Fisher's method), the number of enriched Biological Processes, and the mean Normalized Enrichment Score (NES). We hope the new figure with a representation of the significance of the

functional enrichments can better convey that the gene expression landscape indicating that distinct and evolving transcriptional profiles characterized baseline and on-treatment tumor biopsies from R compared to NR patients. Lesions from R patients showed progressive enrichment for signatures and gene sets revealing activation of adaptive immunity and effective immunomodulation.

*4. Fig 4: For the statement “Several of the genes in the ICR signature showed a progressive expression increase in R patients during therapy, compared to NR patients”: the heatmap visually demonstrates that for the R patients. However, for the NR patients, this is not clearly the case for the associated statement: “By contrast, lesions from NR patients showed significantly higher expression of cell cycle-, EMT- and skin development-related genes”. Moreover, the figure legend states that “Significance values on the right correspond to the difference correspond to all weeks together (p-value from Student's t-Test, ∙: p <0.1, *: p <0.05, **: p < 0.01) (B)”. Student's t-Test is not the appropriate test to compare more than two items, such as all weeks together.*

R: We have updated figure and moved it to Supplementary Fig. 5. We have performed just pairwise tests between time-points and reported the significance.

5. Fig 5A: The following statement requires statistical analysis: “Interestingly, the specific immune subtypes that discriminated against R from NR lesions also showed a progressive increase over time of treatment in R subjects.”

R: We have specified that the increase is a trend but not significant. We have reported the trend as it is suggestive. We are confident that with a higher number of cases, the significance can be reached.

6. Fig 6: It is not clear which data/figure supports the following statement “A rank aggregation analysis of gene expression and gene methylation in NIBIT-M4 tumor biopsies demonstrated the interdependent changes between gene body methylation levels and gene expression induced by guadecitabine”. The statement also mentions “gene body” methylation while the subsequent statements focus on gene promoters; is there a missing link here? Also, in panels B, the axes indicate -log(pvalues) while the pertinent text actually refers to effect sizes (hyper/hypo methylation and up/down regulation) and not to p value significance. It seems the text of the results are not well aligned with the figures or that there are some errors in the labels of the figures, etc...

R: The results previously presented in Figure 6A have been updated with a more focused analysis reported in Figures 6A and 6B and the Supplementary Figure S6A. Where we show the overall methylation pattern across all genomic regions. We report in the revised manuscript a decreasing trend for responders and non-responders (**Figure R2A**, and new Supplementary Figure S6A of the revised manuscript). When we focus on specific genomic regions, the effect of the DHA in the coding part of the genome (exons and intergenic regions) appears to be significantly decreasing in responders but not in NR. Similarly, for promoters, UTRs regions and other regulatory regions such as enhancers and super-enhancers. We also evaluated the specific methylation pattern during therapy in Long interspersed nuclear elements (LINEs), Short interspersed nuclear elements (SINEs), and long terminal repeat (LTR) as they are specifically activated by the demethylating agent. The effect of DHA was a general decreasing trend of methylation

for both R and NR patients (**Figure R2B**, and Figure 6A of the revised manuscript). Therefore, the aggregated rank analysis has been removed because redundant with the new results.

Regarding the Figure 6B of the original manuscript, it has been moved to the supplementary material as Figure S6B, and we have specified how the “starburst plot” is computed. The x and y coordinates are obtained by multiplying the $-\log(\text{p-value})$ of the difference by the sign of the difference in such a way that hypomethylated and upregulated genes lie in the upper right, the hyper-methylated and downregulated genes lie in the lower left quadrant.

7. The last Result section is relatively lengthy. It could benefit from further focus. For eg. some of its statements may better fit in the Discussion, such as “In fact, lesions with defective expression of HLA class I molecules may retain evidence for development of adaptive immunity (High-ICR) and also for CD8+ T cell infiltration, but downmodulation of MHC class I molecules on tumor cells prevents recognition of HLA/neoantigen complexes by T cells, thus suppressing the possibility of immunoediting (therefore the lesions are identified as Non-GIE)”. Also, the part pertinent to Fig 9 may be worth putting as a separate section, focusing on replication of findings.

R: We have accurately updated the paper also following the suggestions of the reviewer.

D. Methods

1. Survival analysis does not address adjustment to important confounders such as patient age, tumor stage, therapy type, etc...

R: Unfortunately, the size of the cohort limits the possibility to perform multivariate analysis. Indeed, the main objective of the current work was to characterize the effect of combined therapy. The NIBIT-M4 is a phase 1b study with primary endpoints being safety, tolerability, and Maximum tolerable dose of treatment; secondary were immune-related disease control rate and objective response rate (ORR) and have been reported elsewhere ²⁴.

2. A section on statistical power is necessary.

R: In the revised manuscript, we have included a section on the statistical power to detect a significant gene expression and methylation difference between R and NR.

E. Discussion:

1. BOR requires abbreviation

R: We have added the definition of Best Overall Response.

2. Some statements may need to be toned down. For eg, the following statement is based on only few rare events: “These two novel findings represent a first selection strategy to increase the success of similar studies”. Another statement is “One of the most interesting messages of our data is that the most evident difference between patients is the dynamic increase of the level of NK-cells and CD8 T-cells in patients that respond to therapy, rather than the tumor microenvironment composition at the baseline.”

I may have missed something here, but the role of the tumor microenvironment cannot be totally ruled out (esp. given the limited sample sizes used for this analysis).

R: In this revised version, several sentences have been revised, including the ones underlined by the reviewer that have been toned down.

Overall:

1. Robustness of the findings: sample size is very limited (as also acknowledged by the authors) especially relative to the high-dimension omics analyses; this drastically limits statistical power. Increasing samples sizes and/or replication in similar trials is crucial. Replication was nicely performed for a portion (but not all) of the results (in Fig 9). A methods section on statistical power is missing. These points need to be addressed collectively.

R: We have included in the revised manuscript a further validation of the guadecitabine-specific signature of 166 genes (**Figure R4** and Figures 3C of the revised manuscript) as further external confirmation of our results. We have also included a section in the paper regarding the statistical power of differential analysis in molecular profiling platforms.

2. UV exposure as a missing link: it seems reasonable to address UV exposure in this study (which seems to be a missing link that can markedly improve the manuscript) especially because: (1) melanoma is largely driven by exposure to UV radiation, (2) the analysed samples already have genetic data that can allow the inference of UV exposure through UV mutational signatures, (3) there have been recent studies highlighting the impact of UV exposure on DNA methylation alterations, mutations, transcriptional regulation and patient prognosis based on melanoma tissues (all of which are data available in this study and which could be directly exploited, and such published studies can be a useful guide for this), and (4) UV exposure is known to associate with TMB and neoantigen load, the latter being one of the interesting aspects addressed in the study. Hence, including UV analysis can link several aspects of this study to one another and enhance the overall interpretation and impact of the study.

R: We computed the COSMIC mutational signature (v3.2)²⁵ frequencies using DeconstructSigs²⁶ for each tumor sample derived from a patient where a matched normal sample was available (n=8).

As expected, UV mutational signatures (SSB7a and SSB7b) were the most frequently observed (**Supplementary Figure S3**). However, there were no observed differences between UV mutational signature rate and response (Chi-squared test p-value = 0.064), mutation load (Chi-squared test p-value = 0.435) and neoantigen load (Chi-squared test p-value = 0.122), respectively.

3. Coherence: the manuscript may benefit from further aligning and focusing of several aspects in relation to one another (abstract, structure of figures, Results, etc...). Some detailed ways to enhance this was provided in my above comments in pertinent sections.

R: We appreciate the time of the reviewer and the detailed suggestions we have tried to fully address. We believe that with the comments of the reviewer together with those by the other reviewers have contributed to significantly improving the original manuscript.

Reviewer #5 (Remarks to the Author): with expertise in immuno-epigenetics, clinical trials

The authors present a 5-year follow-up of the NIBIT-M4 study combining ipilimumab + Guadecitabine in newly diagnosed stage III and IV previously untreated melanoma patients. They previously published results from this study in Clin. Cancer Res. 25, 7351–7362 (2019). They now update their survival data and provide extensive multi-omic correlative analysis performed on serial biopsies from 14 of the 19 patients treated on study. Responses were observed in 6 patients with 3 of these having long term response off treatment and 3 additional patients requiring salvage therapy but with prolonged survival/duration of response. Serial samples are available from all the responders.

The authors have performed an enormous amount of work on these samples. Using multi-omic analysis approaches they have examined changes in global gene expression by whole exome and RNA sequencing as well as methylation using reduced representation bisulfite sequencing. They identified differences in gene signature between responding and non-responding patients to this combined immunotherapy (CTLA4 blockade) in combination with an experimental hypomethylating agent, guadecitabine.

They suggest that the best responses to their therapy were seen in individuals demonstrating both baseline evidence for immune activation and tumor recognition (as characterized by evidence in the tumor of a signature associated with FEWER mutations compared to the number predicted) what they call the genetic immunoediting (GIE) score in combination with the immune constant of rejection score, an established score which describes the number of immune active cells in a tumor environment, thus the combined score provides a sense of both # and function of immune cells within the tested tumors.

Overall, the paper is of significant interest and the authors have produced a wealth of data.

Unfortunately, it is somewhat hard to follow the thread of their story and their takeaway points remain somewhat poorly defined. I would very much like to accept this paper as I think it adds substantially to the field, but some careful editing and more clarity in presentation of the data, assessment of what they have found and in summary of what we should do with their results would be helpful. I have attached a word document with some specific minor suggestions for clarity, but these do not address the main problem with this paper, which is a lack of clear message.

This issue could be readily addressed by edits to the discussion and results sections to provide the reader with a summary of what they have found in each figure and how they put these data together for the field.

R: we are flattered by the positive comments of the reviewer. We have revised the paper according to his suggestions as reported below.

Minor Comments

Q: This paper uses excessive acronyms many of which are similar (ICI, ICR, etc) which make readership challenging for the non-cognoscenti.

R: We have extensively reviewed the manuscript trying to avoid as much as possible acronyms.

Q: Page 5 line 109: please define DCR.

R: We thank the reviewer for this comment that allowed us to identify a typo in the submitted version of the manuscript since DCR was meant to be DC (Page 5 line 109), we have amended the manuscript accordingly. Also, based on the reviewer's suggestion we have defined DC [i.e., CR, Partial Response (PR), or Stable Disease (SD)] in the section Materials and Methods of the revised version of the manuscript.

Q: Consider changing the acronym used for DNA hypomethylating agents from DHA to either DNA methyltransferase inhibitors (DNMTi) or hypomethylating agents (HMAs) these abbreviations are perhaps used more commonly within the field and might be easier to follow. As the authors are no doubt aware, these drugs hypomethylate RNA as well as DNA and also cause DNA damage.

R: Thank you for your detailed comment. We have amended the manuscript referring DHA as HMAs.

Q: Responding (R) and non-responding (NR) patients are abbreviated as indicated here. In figure 6 R is also used to indicate spearman correlation coefficients rho. I would suggest not conflating these two abbreviations. The R and NR are defined in the figure legends, but not in the manuscript text the first time these abbreviations come up on page 5 line 112.

R: In agreement with the reviewer's suggestion, we have defined R and NR in the text of the revised version of the manuscript the first time these abbreviations appear (page 5). Also, we have used the lowercase "r" to indicate the correlation coefficient.

Queries:

Q: Page 13 lines 357-361: The authors discuss that their work identified specific melanomas which appear to have downregulated or developed defects in antigen processing and presentation characterized by loss of MHC class 1. These authors and others in the field have demonstrated that malignant cells treated with DNA methyltransferase inhibitors can upregulate expression of MHC proteins. They do not clearly discuss this in the results section. One might hypothesize based upon these prior data that the DNA methyltransferase inhibitors might improve the response to immune checkpoint therapy most dramatically in melanomas with high ICR scores but lower evidence of immune editing, and moreover, serial monitoring of these relative scores might show increased expression of MHC class 1 in the DNMTi treated samples. ADDENDUM: Figure 4A page 24 line 607 seems to suggest that among responders there was upregulation in HLA class 1 and 2 expression, while non-responders did not seem to demonstrate this change as clearly, particularly for class 2 expression which appears to be uniformly down in the non-responders. Please comment on this in the results and discussion section as this seems to be a very important finding.

R: Thank you for suggesting to deepen these important points. We have commented on the upregulation of HLA molecules in the Results and in the Discussion.

Q: Results page 5 lines 104-110 and Page 14: the original study NIBIT-M4 enrolled pre-treated or untreated patients with melanoma to receive guadecitabine + ipilimumab. They had 6 long term survivors with 3 long term CRs who were off study and 5 individuals with durable CRs. Among the 3 who recurred but were

not alive, what were the subsequent lines of therapy? Did any of these patients receive 2nd generation ICI therapy and did they have a better than expected response? (I see these data in the figure legend, but it is not discussed in the manuscript text, I would briefly mention it there) and it would be nice to delineate which of the patients received which therapy on the swimmer's plot.

R: We thank the reviewer for this relevant question. To better inform the readers we have revised Figure 1 that now includes subsequent treatment(s) (if any) of each patient enrolled in the study. Among the 3 patients who recurred and received subsequent line(s) of therapy but were not alive, no better than expected clinical responses were observed in the single patient who received 2nd generation ICI. As suggested, we have briefly mentioned the subsequent treatments in the revised manuscript text.

Q: Page 14: you have RRBS on all the serial samples. What did this show? In the tumors with mutation in DNMT1 and SETD2 show hypomethylation of the tumors at baseline without evidence of an effect from the DNMTi to support the authors hypothesis they suggest for resistance? These are single patient series, but might be hypothesis generating.

R: We have included in the revised manuscript more details analysis of the overall methylation profiles in R and NR cohorts. The overall methylation pattern across all genomic regions, shows a decreasing trend for responders and non-responders (**Figure R2A** and Supplementary Figure S6A of the revised manuscript).

We also considered the reviewer's suggestion to explore the effect of the mutation of DNMT1 and SETD2 at the methylation level. Three patients, all non-responders, harbored mutations in these two genes. For this purpose, we evaluated the evolution of methylation under therapy. The increasing or decreasing trend was evaluated on the basis of the inclination of the linear regression line between the three time points. Interestingly, mutant samples did not show the same decreasing pattern that we observed in the wild-type lesions. We analyzed all genomic regions, and then specifically the coding, intergenic, intronic regions, and regulatory regions. The decreasing methylation pattern was observed in the wild-type and not in the mutant lesions (**Figure R8** and Supplementary Figure S2A and S2B of the revised manuscript). This was also confirmed in long terminal repeat (LTR) including endogenous retroviral elements (ERVs) which are particularly important for the immune response to tumors, even in the context of ICB¹³. Previous studies have shown that inducing the expression of ERV sequences, demethylating agents can activate the viral mimicry response secondary to intracellular recognition of viral dsRNA^{14,15}. This response explains the promotion of type I IFN and innate immunity pathways that characterize the guadecitabine-specific gene signature recently defined by us in melanoma cells¹¹. Therefore, even if supported by three cases, these results further confirm the action of the demethylating agent and that defects in its targets could prevent the efficacy of the combined therapy, also suggesting patient selection strategies for enrolment in future clinical studies.

Figure R8. Evolution of the methylation pattern for lesions harboring mutations in chromatin organization (*DMNT1*, *SETD2*) and wild-type. Genomic regions (A), Regulatory Regions (B), and Endogenous Retroviruses (C).

Q: Page 23 Figure 3A: I presume that the responders in 3A are in peach/orange and the non-responders are in green as listed in figure 3B but I would like this to be spelled out in the figure legend please. I don't really understand or follow what is being presented in this figure 3A. What does normalized enrichment score mean? Does the spread from baseline to on treatment indicated increased transcriptional activity? I find this figure very difficult to follow.

R: This figure has been criticized also by other reviewers, and we acknowledge that its interpretation was difficult. We have re-analyzed the data and generated a new Figure 3A which is reported here in **Figure R5**. It now contains more quantitative information and we hope that the message conveyed by the figure is more clear.

Q: Page 25 figure 5: I "Immune Context" might be clearer. Contexture is defined by Merriam Webster as the act, process, or manner of weaving parts into a whole or a structure so formed. The use of this description is not incorrect, but it is an unfamiliar word and I'm not sure that the authors have completely finished their 'weaving'.

R: We have changed the title of the figure to “Immune Microenvironment”. The reviewer’s comment is well-noticed.

Q: Figure 9: I don’t think the authors looked independently at 3 different cohorts. I believe they mean to say that they examined ICR/GIE scores from 83 patients compiled from 3 separate studies and considered them as a group against which to validate their score. This needs to be clarified in the figure legend, it is described correctly in the text.

R: Thank you for noticing this. We have changed the figure title with “Validation of the ICR/GIE score in patients from different cohorts”.

References

1. Roulois, D. *et al.* DNA-Demethylating Agents Target Colorectal Cancer Cells by Inducing Viral Mimicry by Endogenous Transcripts. *Cell* **162**, 961–973 (2015).
2. Johnson, D. B. *et al.* Impact of NRAS mutations for patients with advanced melanoma treated with immune therapies. *Cancer Immunol. Res.* **3**, 288–295 (2015).
3. Mangana, J. *et al.* Analysis of BRAF and NRAS Mutation Status in Advanced Melanoma Patients Treated with Anti-CTLA-4 Antibodies: Association with Overall Survival? *PLoS ONE* **10**, e0139438 (2015).
4. Wang, M., Yu, L., Wei, X. & Wei, Y. Role of tumor gene mutations in treatment response to immune checkpoint blockades. *Precis. Clin. Med.* **2**, 100–109 (2019).
5. Larkin, J. *et al.* Efficacy and Safety of Nivolumab in Patients With BRAF V600 Mutant and BRAF Wild-Type Advanced Melanoma: A Pooled Analysis of 4 Clinical Trials. *JAMA Oncol.* **1**, 433–440 (2015).
6. Rossfeld, K. *et al.* Metastatic melanoma patients' sensitivity to ipilimumab cannot be predicted by tumor characteristics. *Int J Surg Oncol (N Y)* **2**, e43 (2017).
7. Adib, E. *et al.* CDKN2A alterations and response to immunotherapy in solid tumors. *Clin. Cancer Res.* **27**, 4025–4035 (2021).
8. Horn, S. *et al.* Tumor CDKN2A-Associated JAK2 Loss and Susceptibility to Immunotherapy Resistance. *J Natl Cancer Inst* **110**, 677–681 (2018).
9. Zhang, Z. *et al.* EPHA7 mutation as a predictive biomarker for immune checkpoint inhibitors in multiple cancers. *BMC Med.* **19**, 26 (2021).
10. Celus, W. *et al.* Plexin-A4 Mediates Cytotoxic T-cell Trafficking and Exclusion in Cancer. *Cancer Immunol. Res.* **10**, 126–141 (2022).
11. Anichini, A. *et al.* Landscape of immune-related signatures induced by targeting of different epigenetic regulators in melanoma: implications for immunotherapy. *J. Exp. Clin. Cancer Res.* **41**, 325 (2022).
12. Shah, N. M. *et al.* Pan-cancer analysis identifies tumor-specific antigens derived from transposable elements. *Nat. Genet.* **55**, 631–639 (2023).
13. Ng, K. W. *et al.* Antibodies against endogenous retroviruses promote lung cancer immunotherapy. *Nature* **616**, 563–573 (2023).

14. Loo Yau, H., Ettayebi, I. & De Carvalho, D. D. The cancer epigenome: exploiting its vulnerabilities for immunotherapy. *Trends Cell Biol.* **29**, 31–43 (2019).
15. Chen, R., Ishak, C. A. & De Carvalho, D. D. Endogenous retroelements and the viral mimicry response in cancer therapy and cellular homeostasis. *Cancer Discov.* **11**, 2707–2725 (2021).
16. Wijetunga, N. A. *et al.* SMITE: an R/Bioconductor package that identifies network modules by integrating genomic and epigenomic information. *BMC Bioinformatics* **18**, 41 (2017).
17. Chen, S. *et al.* Epigenetic priming enhances antitumor immunity in platinum-resistant ovarian cancer. *J. Clin. Invest.* **132**, (2022).
18. Papadatos-Pastos, D. *et al.* Phase 1, dose-escalation study of guadecitabine (SGI-110) in combination with pembrolizumab in patients with solid tumors. *J. Immunother. Cancer* **10**, (2022).
19. Frattini, V. *et al.* A metabolic function of FGFR3-TACC3 gene fusions in cancer. *Nature* **553**, 222–227 (2018).
20. Roelands, J. *et al.* An integrated tumor, immune and microbiome atlas of colon cancer. *Nat. Med.* (2023) doi:10.1038/s41591-023-02324-5.
21. Angelova, M. *et al.* Evolution of Metastases in Space and Time under Immune Selection. *Cell* **175**, 751-765.e16 (2018).
22. Bruni, D., Angell, H. K. & Galon, J. The immune contexture and Immunoscore in cancer prognosis and therapeutic efficacy. *Nat. Rev. Cancer* **20**, 662–680 (2020).
23. Rooney, M. S., Shukla, S. A., Wu, C. J., Getz, G. & Hacohen, N. Molecular and genetic properties of tumors associated with local immune cytolytic activity. *Cell* **160**, 48–61 (2015).
24. Di Giacomo, A. M. *et al.* Guadecitabine Plus Ipilimumab in Unresectable Melanoma: The NIBIT-M4 Clinical Trial. *Clin. Cancer Res.* **25**, 7351–7362 (2019).
25. Alexandrov, L. B. *et al.* The repertoire of mutational signatures in human cancer. *Nature* **578**, 94–101 (2020).
26. Rosenthal, R., McGranahan, N., Herrero, J., Taylor, B. S. & Swanton, C. DeconstructSigs: delineating mutational processes in single tumors distinguishes DNA repair deficiencies and patterns of carcinoma evolution. *Genome Biol.* **17**, 31 (2016).

REVIEWERS' COMMENTS

Reviewer #1 (Remarks to the Author):

I am grateful for the authors' extensive revisions of their work and their thorough clarifications! I only have one minor comment:

“Many thanks for raising this point that allows us to better clarify how we segregate the samples into the four groups GIE/High-ICR non-GIE/ICR, GIE/low-ICR, and no-GIE/low-ICR. Therefore, we use this value to nominate high-ICR versus low-ICR. Due to the low number of samples, we found this option to be the most natural, robust, and non-arbitrary choice.” I understand how the classification is done by the authors. I think this classification is ok, and the authors just need to explain it clearly in their manuscript. In their current manuscript, this has not been explained clearly anywhere, unless I missed something?

Reviewer #2 (Remarks to the Author):

With the revisions, the manuscript is substantially improved. All my previous questions were addressed appropriately

Reviewer #3 (Remarks to the Author):

The authors have addressed my comments.

In terms of the purity analysis and effect on determining the relative contribution of gene expression programs I understand the authors hesitation to include data that do not support drawing a conclusion either way. It would be worthwhile to mention the need to account for the cellular composition at different time points for inferring changes in gene expression programs in the the discussion.

Reviewer #5 (Remarks to the Author):

The authors have adequately addressed my concerns. The revised manuscript is clear, provides a hypothesis and an excellent explanation of the results they found. The revised figures are significantly improved and the letter very clearly addressed the concerns raised by reviewers.

Reply to Reviewer's comments of the paper

Guadecitabine plus ipilimumab in unresectable melanoma: five-year follow-up and integrated multi-omic analysis in the phase 1b NIBIT-M4 trial

We are happy that the reviewers acknowledge that most of their comments have been properly addressed by us. Please find below the replies to Reviewers #1 and #3. We have incorporated these revisions in the current version of the manuscript.

Reviewer #1 (Remarks to the Author)

I am grateful for the authors' extensive revisions of their work and their thorough clarifications! I only have one minor comment:

"Many thanks for raising this point that allows us to better clarify how we segregate the samples into the four groups GIE/High-ICR non-GIE/ICR, GIE/low-ICR, and no-GIE/low-ICR. Therefore, we use this value to nominate high-ICR versus low-ICR. Due to the low number of samples, we found this option to be the most natural, robust, and non-arbitrary choice." I understand how the classification is done by the authors. I think this classification is ok, and the authors just need to explain it clearly in their manuscript. In their current manuscript, this has not been explained clearly anywhere, unless I missed something?

R: We reported the adopted thresholds for stratification of patients into the four GIE/ICR classes in lines 662-664 of the previous manuscript. We updated this paragraph with the arguments reported in the rebuttal used to justify the threshold of the GIE score and the Network Enrichment Score (NES) of the ICR signature. In particular, we included the following paragraph:

"Since the GIE is a ratio between observed and expected neo-antigens, we used "GIE" or "non-GIE" definitions to nominate samples with $GIE < 1$ or $GIE > 1$, respectively. Analogously, we adopted the normalized enrichment score (NES) to evaluate the activation of the ICR signature at the single-sample level using our yaGST tool. A value of $NES > 0$ means positive activation, whereas a value of $NES < 0$ means a significant negative activation. Accordingly, these two conditions were nominated "high-ICR" versus "low-ICR", respectively.

Reviewer #3 (Remarks to the Author)

The authors have addressed my comments.

In terms of the purity analysis and effect on determining the relative contribution of gene expression programs I understand the authors hesitation to include data that do not support drawing a conclusion either way. It would be worthwhile to mention the need to account for the

cellular composition at different time points for inferring changes in gene expression programs in the discussion.

R: We report these considerations in the Discussion of the updated manuscript. In particular, we state that: “The inference of changes in gene expression programs needs to account for the cellular composition at different time points. The actual profiling platform, with half of the cases lacking the normal reference, are not ideal for an accurate estimation of the purity. Single-cell profiling will be performed on this cohort in future studies to address this point.”